# BROKEN NEURAL SCALING LAWS

**Ethan Caballero**
Mila, McGill University
`ethan.victor.caballero@gmail.com`
`ethan.caballero@mila.quebec`

**Kshitij Gupta**
Mila, University of Montreal

**Irina Rish**
Mila, University of Montreal

**David Krueger**
University of Cambridge

## ABSTRACT

We present a smoothly broken power law functional form (referred to by us as a *broken neural scaling law* (BNSL)) that accurately models and extrapolates the scaling behaviors of deep neural networks (i.e. how the evaluation metric of interest varies as the amount of compute used for training, number of model parameters, training dataset size, or upstream performance varies) for various architectures and for each of various tasks within a large and diverse set of upstream and downstream tasks, in zero-shot, prompted, and fine-tuned settings. This set includes large-scale vision, language, audio, video, diffusion, generative modeling, multimodal learning, contrastive learning, AI alignment, robotics, out-of-distribution (OOD) generalization, continual learning, uncertainty estimation / calibration, out-of-distribution detection, adversarial robustness, molecules, computer programming/coding, math word problems, arithmetic, unsupervised/self-supervised learning, and reinforcement learning (single agent and multi-agent). When compared to other functional forms for neural scaling behavior, this functional form yields extrapolations of scaling behavior that are considerably more accurate on this set. Moreover, this functional form accurately models and extrapolates scaling behavior that other functional forms are incapable of expressing such as the non-monotonic transitions present in the scaling behavior of phenomena such as double descent and the delayed, sharp inflection points present in the scaling behavior of tasks such as arithmetic. Lastly, we use this functional form to glean insights about the limit of the predictability of scaling behavior. See arXiv for longer version of this paper. Code is available at `https://github.com/ethancaballero/broken_neural_scaling_laws`

## 1 INTRODUCTION

The amount of compute used for training, number of model parameters, and training dataset size of the most capable artificial neural networks keeps increasing and will probably keep rapidly increasing for the foreseeable future. However, no organization currently has direct access to these larger resources of the future; and it has been empirically verified many times that methods which perform best at smaller scales often are no longer the best performing methods at larger scales (e.g., one of such examples can be seen in Figure 2 (right) of Tolstikhin et al. (2021)). To work on, identify, and steer the methods that are most probable to stand the test-of-time as these larger resources come online, one needs a way to predict how all relevant performance evaluation metrics of artificial neural networks vary in all relevant settings as scale increases.

Neural scaling laws (Cortes et al., 1994; Hestness et al., 2017; Rosenfeld et al., 2019; Kaplan et al., 2020; Zhai et al., 2021; Abnar et al., 2021; Alabdulmohsin et al., 2022; Brown et al., 2020) aim to predict the behavior of large-scale models from smaller, cheaper experiments, allowing to focus on the best-scaling architectures, algorithms, datasets, and so on. The upstream/in-distribution test loss typically (but not always!) falls off as a power law with increasing data, model size and compute. However, the downstream/out-of-distribution performance, and other evaluation metrics of interest (even upstream/in-distribution evaluation metrics) are often less predictable, sometimes exhibiting inflection points (on a linear-linear plot) and non-monotonic behaviors. Discovering *universal scaling laws* that accurately model a wide range of potentially unexpected behaviors is clearly important not only for identifying that which scales best, but also for AI safety, as predicting the emergence of novel capabilities at scale could prove crucial to responsibly developing and deploying increasingly advanced AI systems. The functional forms of scaling laws evaluated in previous work are not up to this challenge.

One salient defect is that they can only represent monotonic functions. They thus fail to model the striking phenomena of double-descent (Nakkiran et al., 2021), where increased scale temporarily decreases test performance before ultimately leading to further improvements. Many also lack the expressive power to model inflection points (on a linear-linear plot), which can be observed empirically for many downstream tasks, and even some upstream tasks, such as our $N$-digit arithmetic task, or the modular arithmetic task introduced by Power et al. (2022) in their work on "grokking".

To overcome the above limitations, we present *broken neural scaling laws (BNSL)* - a functional form that generalizes power laws (linear in log-log plot) to "smoothly broken" power laws, i.e. a smoothly connected piecewise (approximately) linear function in a log-log plot. An extensive empirical evaluation demonstrates that BNSL accurately model and extrapolate the scaling behaviors for various architectures and for each of various tasks within a large and diverse set of upstream and downstream tasks, in zero-shot, prompted, and fine-tuned settings. This set includes large-scale vision, language, audio, video, diffusion, generative modeling, multimodal learning, contrastive learning, AI alignment, robotics, out-of-distribution generalization, continual learning, uncertainty estimation / calibration, out-of-distribution detection, adversarial robustness, molecules, computer programming/coding, math word problems, arithmetic, unsupervised/self-supervised learning, and reinforcement learning (single agent and multi-agent). When compared to other functional forms for neural scaling behavior, this functional form yields extrapolations of scaling behavior that are considerably more accurate on this set. It captures well the non-monotonic transitions present in the scaling behavior of phenomena such as double descent and the delayed, sharp inflection points present in the scaling behavior of tasks such as arithmetic.

## 2 THE FUNCTIONAL FORM OF BROKEN NEURAL SCALING LAWS

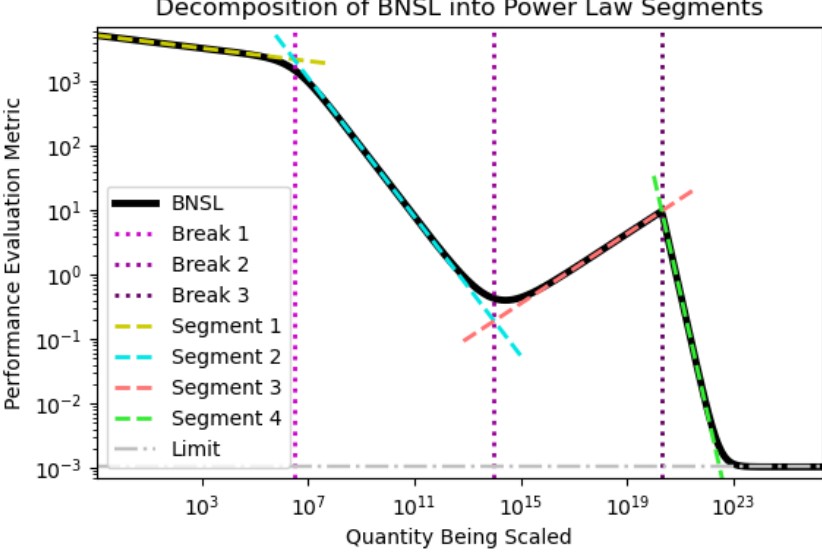

Figure 1: A Broken Neural Scaling Law (BNSL) (dark black solid line) (with 3 breaks where purple dotted lines intersect with dark black solid line) that contains 4 individual power law segments (where the dashed lines that are yellow, blue, red, and green overlay the dark black solid line). The 1st and 2nd break are very smooth; the 3rd break is very sharp. See Section 2 for more details.

The general functional form of a broken neural scaling law (BNSL) is given as follows:

$$y = a + \left( b x^{-c_0} \right) \prod_{i=1}^{n} \left( 1 + \left( \frac{x}{d_i} \right)^{1/f_i} \right)^{-c_i * f_i}, \tag{1}$$

where $y$ represents the performance evaluation metric (e.g. prediction error, cross entropy, calibration error, AUROC, BLEU score percentage, reward, Elo rating, or FID score) (**downstream or upstream**) and $x$ represents a quantity that is being scaled (e.g. number of model parameters, amount of compute used for training, training dataset size, or upstream performance). The remaining parameters $a, b, c_0, c_1...c_n, d_1...d_n, f_1...f_n$ are unknown constants that must be estimated by fitting the above functional form to the $(x, y)$ data points. (In our experiments, SciPy curve-fitting library (Virtanen et al., 2020) was used.)

The constants in equation 1 are interpreted as follows. Constant $n$ represents the number of (smooth) "breaks" (i.e. transitions) between $n+1$ consecutive approximately linear (on a log-log plot) segments, for a total of $n+1$ approximately linear segments (on a log-log plot); when $n = 0$, equation 1 becomes $y = a + bx^{-c_0}$. Constant $a$ represents the limit as to how far the value of $y$ (performance evaluation metric) can be reduced (or maximized) even if $x$ (the quantity being scaled) goes to infinity. Constant $b$ represents the offset of functional form on a log-log plot (analogous to the intercept $b$ in $y = mx + b$ on a linear-linear plot). Constant $c_0$ represents the slope of the first approximately linear region on a log-log plot. Constant $c_i$ represents the difference in slope of the $(i)$th approximately linear region and $(i+1)$th approximately linear region on a log-log plot. Constant $d_i$ represents where on the x-axis the break between the $(i)$th and the $(i+1)$th approximately linear region (on a log-log plot) occurs. Constant $f_i$ represents the sharpness of break between the $(i)$th and the $(i+1)$th approximately linear region on a log-log plot; smaller (nonnegative) values of $f_i$ yield a sharper break and intervals (before and after the $(i)$th break) that are more linear on a log-log plot; larger values of $f_i$ yield a smoother break and intervals (before and after the $(i)$th break) that are less linear on a log-log plot.

For mathematical analysis and explanation of why Equation 1 is smoothly piece-wise (approximately) linear function on a log-log plot, see Appendix A.1. For mathematical decomposition of Equation 1 into the power law segments it is composed of (e.g. as in Figure 1), see Appendix A.2.

Note that, while an intuition for using such smoothly connected approximately piece-wise linear (in log-log plot) function was that, with enough segments, it could fit well any smooth univariate scaling function, it remained unclear whether BNSL would also *extrapolate* well; yet as we demonstrate below, it extrapolates quite accurately. Additionally, we find that the number of breaks needed to accurately model an entire scaling behavior is often quite small.

## 3 RELATED WORK

To the best of our knowledge, Cortes et al. (1994) was the first paper to model the scaling of multi-layer neural network's performance as a power law (also known as a scaling law) (plus a constant) of the form $y = ax^b + c$ in which $x$ refers to training dataset size and $y$ refers to test error; we refer to that functional form as M2. Hestness et al. (2017) showed that the functional form, M2, holds over many orders of magnitude. Rosenfeld et al. (2019) demonstrated that the same functional form, M2, applies when $x$ refers to model size (number of parameters). Kaplan et al. (2020) brought "neural" scaling laws to the mainstream and demonstrated that the same functional form, M2, applies when $x$ refers to the amount of compute used for training. Abnar et al. (2021) proposed to use the same functional form, M2, to relate downstream performance to upstream performance. Zhai et al. (2021) introduced the functional form $y = a(x + d)^b + c$, (referred to by us as M3) where d represents the scale at which the performance starts to improve beyond the random guess loss (a constant) and transitions to a power law scaling regime. Alabdulmohsin et al. (2022) proposed functional form $(y - \epsilon_\infty)/((\epsilon_0 - y)^a) = bx^c$, (referred to by us as M4) where $\epsilon_\infty$ is irreducible entropy of the data distribution and $\epsilon_0$ is random guess performance, for relating scale to performance and released a scaling laws benchmark dataset that we use in our experiments.

Hernandez et al. (2021) described a smoothly broken power law functional form (consisting of 5 constants after reducing redundant variables) in equation 6.1 of their paper, when relating scale and downstream performance. While this functional form can be summed with an additional constant representing unimprovable performance to obtain a functional form whose expressivity is equivalent to our BNSL with a single break, it is important to note that (i) Hernandez et al. (2021) describes this form only in the specific context, when exploring how fine-tuning combined with transfer learning scales as a function of the model size - thus, their functional form contains a break only with respect to number of model parameters but not with respect to other input quantities which we do explore such as dataset size, amount of compute, and upstream performance; (ii) Hernandez et al. (2021) mentioned this equation in passing and as a result did not try to fit or verify this functional form on any data; (iii) they arrived at it simply via combining the scaling law for transfer (that was the focus of their work) with a scaling law for pretraining data; (iv) they did not identify it as a smoothly broken power law, or note any qualitative advantages of this functional form; (v) they did not discuss the family of functional forms with multiple breaks.

Finally, we would like to mention that smoothly broken power law functional forms, equivalent to equation 1, are commonly used in the astrophysics literature (e.g. dam (2017)) as they happen to model well a variety of physical phenomena. This inspired us to investigate their applicability to a wide range of deep neural scaling phenomena as well.

## 4 Theoretical Limitations of Previously Proposed Scaling Laws

Our use of BNSLs is inspired by the observation that scaling is not always well predicted by a simple power law; nor are many of the modifications which have been applied in previous works sufficient to capture the qualitative properties of empirical scaling curves. Here we show mathematically two qualitative defects of these functional forms:

1. They are strictly monotonic (first-order derivative does not change its sign) and thus unable to fit double descent phenomena.

2. They cannot express inflection points (second-order derivative does not change its sign), which are frequently observed empirically. An exception to this is M4, proposed by Alab-dulmohsin et al. (2022).

Note that these functional forms *can* exhibit inflection points on the log-log axes which are commonly used for plotting scaling data (as it was observed in several prior works). However, for inflection points on a *linear-linear* plot, the extra expressiveness of broken neural scaling laws appears to be necessary (and sufficient). Figure 3 and Figure 4, provide examples of BNSLs producing non-monotonic behavior and inflection points, respectively, establishing the capacity of this functional form to model these phenomena that occur in real scaling behavior.

| name | $f(x)$ | $f'(x)$ | $f''(x)$ |
|------|--------|---------|----------|
| M1 | $ax^b$ | $abx^{b-1}$ | $ab(b-1)x^{b-2}$ |
| M2 | $ax^b + c$ | $abx^{b-1}$ | $ab(b-1)x^{b-2}$ |
| M3 | $a(x^{-1} + d)^{-b} + c$ | $\frac{ab}{x(1+dx)(d+1/x)^b}$ | $abx^{(b-2)}(1+dx)^{(-2-b)}(b-1-2dx)$ |

Table 1: Previously proposed functional forms M1, M2, M3 and their (first and second order) derivatives. See Equation 2 for M4.

**M1, M2, M3 functional forms cannot model non-monotonic behavior or inflection points:** First, recall that expressions of the form $m^n$ can only take the value 0 if $m = 0$. We now examine the expressions for the first and second derivatives of M1, M2, M3, provided in Table 1, and observe that they are all continuous and do not have roots over the relevant ranges of their variables, i.e. $x > 0$ in general and $b < 0$ in the case of M3 (we require $x > 0$ because model size, dataset size, and compute are always non-negative). This implies that, for any valid settings of the parameters $a, b, c, d, x$, these functional forms are monotonic (as the first derivative never changes sign), and that they lack inflection points (since an inflection point must have $f''(x) = 0$).

**M4 functional form cannot model non-monotonic behavior.** The case of M4 is a bit different, since the relationship between $y$ and $x$ in this case is expressed as an inverse function, i.e.

$$x = g(y) = \left( \frac{y - \epsilon_\infty}{b(\epsilon_0 - y)^a} \right)^{1/c} \tag{2}$$

However, non-monotonicity of $y$ as an inverse function $y = g^{-1}(x)$ is ruled out, since that would imply two different values of $x = g(y)$ can be obtained for the single value of $y$ – this is impossible, since $f(y)$ maps each $y$ deterministically to a single value of $x$. As a result, M4 cannot express non-monotonic functions.

**M4 functional form can model inflection points.** It is easy to see that if $y = g^{-1}(x)$ had an inflection point, then $x = g(y)$ would have it as well. This is because an inflection point is defined as a point $x$ where $f(x)$ changes from concave to convex, which implies that $g(y)$ changes from convex to concave, since the inverse of a convex function is concave; the root(s) of $g''(y)$ are the point(s) at which this change occurs. Using Wolfram Alpha[1] and matplotlib (Hunter, 2007), we observe that M4 is able to express inflection points, e.g. $(a, b, c, \epsilon_0, \epsilon_\infty, x, y) = (1, 1, -2, 3/4, 1/4, 1/\sqrt{3}, 5/8)$, or $(a, b, c, \epsilon_0, \epsilon_\infty, x, y) = (2, 1, -3, 2/3, 1/3, (-5/6 + \sqrt{3}/2)^{1/3}, 1/\sqrt{3})$.

---

[1] https://www.wolframalpha.com/

## 5 EMPIRICAL RESULTS: FITS AND EXTRAPOLATIONS OF FUNCTIONAL FORMS

We now show the fits and extrapolations of various functional forms. **In all plots here and onward and in the appendix, black points are points used for fitting a functional form, green points are the held-out points used for evaluating extrapolation of functional form fit to the black points, and a red line is the BNSL that has been fit to black points. 100% of the plots in this paper here and onward and in the appendix contain green point(s) for evaluating extrapolation.** Please refer to Appendix Section A.6 for further experimental details on fitting BNSL.

**Except when stated otherwise**, each plot contains a single break of a BNSL fit to black points that are smaller (along the x-axis) than the green points.

All the extrapolation evaluations reported in the tables are reported in terms of root mean squared log error (RMSLE) ± root standard log error. See Appendix A.3 for definition of RMSLE and Appendix A.4 for definition of root standard log error.

| Domain | M1 ↑ | M2 ↑ | M3 ↑ | M4 ↑ | BNSL ↑ |
|---|---|---|---|---|---|
| Downstream Image Classification | 2.78% | 4.17% | 9.72% | 13.89% | **69.44%** |
| Language | 10% | 5% | 10% | 0% | **75%** |

Table 2: Percentage of tasks by domain where each functional form is the best for extrapolation of scaling behavior. Numbers for M1, M2, M3, and M4 were obtained via correspondence with authors of Alabdulmohsin et al. (2022). See Sections 5.1 and 5.2 for more details.

### 5.1 VISION

Using the scaling laws benchmark of Alabdulmohsin et al. (2022), we evaluate how well various functional forms extrapolate performance on vision tasks as training dataset size increases. In this large-scale vision subset of the benchmark, the tasks that are evaluated are error rate on each of various few-shot downstream image classification (IC) tasks; the downstream tasks are: Birds 200 (Welinder et al., 2010), Caltech101 (Fei-Fei et al., 2004), CIFAR-100 (Krizhevsky et al., 2009), and ImageNet (Deng et al., 2009). The following architectures of various sizes are pretrained on subsets of JFT-300M (Sun et al., 2017): big-transfer residual neural networks (BiT) (Kolesnikov et al., 2020), MLP mixers (MiX) (Tolstikhin et al., 2021), and vision transformers (ViT) (Dosovitskiy et al., 2020). As can be seen in Tables 2 and 3, BNSL yields extrapolations with the lowest RMSLE (Root Mean Squared Logarithmic Error) for 69.44% of tasks of any of the functional forms, while the next best functional form performs the best on only 13.89% of the tasks.

To view plots of BNSL on each of these tasks, see figures 24, 25, 26, 30 in Appendix A.26. To view plots of M1, M2, M3, M4 on each of these tasks, see Appendix A.4 of Alabdulmohsin et al. (2022).

In Section A.8, we additionally show that BNSL yields accurate extrapolations of performance on large-scale downstream vision tasks when amount of compute used for (pre-)training is on the x-axis and compute is scaled in the manner that is Pareto optimal with respect to the performance evaluation metric on the y-axis (downstream accuracy in this case).

In Section A.9, we additionally show that BNSL yields accurate extrapolations of the scaling behavior of diffusion generative models of images when amount of compute used for (pre-)training is on the x-axis and compute is scaled in the manner that is Pareto optimal with respect to the performance evaluation metric on the y-axis (NLL and FID score in this case).

In Section A.10, we additionally show that BNSL yields accurate extrapolations of the scaling behavior of generative models of video.

In Section A.20, we show that BNSL yields accurate extrapolations of robotics scaling behavior (out-of-distribution generalization and in-distribution generalization).

In Section A.19, BNSL accurately extrapolates the scaling behavior of continual learning.

In Section A.15, BNSL accurately extrapolates the scaling behavior of adversarial robustness.

In Section A.24, we show that BNSL accurately extrapolates the scaling behavior of the downstream performance of multimodal contrastive learning (i.e. non-generative unsupervised learning).

In Section A.11, we additionally show that BNSL yields accurate extrapolations of the scaling behavior when data is pruned Pareto optimally (such that each point along the x-axis uses the subset of the dataset that yields the best performance (y-axis value) for that dataset size (x-axis value)).

In Section A.12, we additionally show that BNSL yields accurate extrapolations when upstream performance is on the x-axis and downstream performance is on the y-axis.

In Section A.7, we additionally show that BNSL accurately extrapolates to scales that are an **order of magnitude** larger than the maximum (along the x-axis) of the points used for fitting.

| Task | Model | M1 ↓ | M2 ↓ | M3 ↓ | M4 ↓ | BNSL ↓ |
|---|---|---|---|---|---|---|
| Birds 200 10-shot | BiT/101/3 | 9.13e-2 ± 2.8e-3 | 9.13e-2 ± 2.8e-3 | 9.13e-2 ± 2.8e-3 | 2.95e-2 ± 1.3e-3 | **1.76e-2 ± 1.1e-3** |
| Birds 200 10-shot | BiT/50/1 | 6.88e-2 ± 7.5e-4 | 6.88e-2 ± 7.5e-4 | 5.24e-2 ± 6.2e-4 | 2.66e-2 ± 5.3e-4 | **1.19e-2 ± 3.5e-4** |
| Birds 200 10-shot | MiX/B/16 | 9.15e-2 ± 1.1e-3 | 9.15e-2 ± 1.1e-3 | 3.95e-2 ± 7.0e-4 | 4.62e-2 ± 8.2e-4 | **3.04e-2 ± 6.9e-4** |
| Birds 200 10-shot | MiX/L/16 | 5.51e-2 ± 1.4e-3 | 5.51e-2 ± 1.4e-3 | 5.51e-2 ± 1.4e-3 | 5.15e-2 ± 1.7e-3 | **1.85e-2 ± 8.9e-4** |
| Birds 200 10-shot | ViT/B/16 | 6.77e-2 ± 1.1e-3 | 6.77e-2 ± 1.1e-3 | 3.52e-2 ± 8.1e-4 | **1.51e-2 ± 6.2e-4** | 1.69e-2 ± 7.0e-4 |
| Birds 200 10-shot | ViT/S/16 | 3.95e-2 ± 1.2e-3 | 3.95e-2 ± 1.2e-3 | 3.74e-2 ± 1.1e-3 | 1.85e-2 ± 7.9e-4 | **1.09e-2 ± 6.1e-4** |
| Birds 200 25-shot | BiT/101/3 | 9.41e-2 ± 3.2e-3 | 9.41e-2 ± 3.2e-3 | 9.41e-2 ± 3.2e-3 | 6.38e-2 ± 2.0e-3 | **1.55e-2 ± 1.3e-3** |
| Birds 200 25-shot | BiT/50/1 | 1.10e-1 ± 1.0e-3 | 7.29e-2 ± 8.0e-4 | 1.52e-2 ± 4.9e-4 | 1.97e-2 ± 5.6e-4 | **1.33e-2 ± 4.4e-4** |
| Birds 200 25-shot | MiX/B/16 | 1.40e-1 ± 1.9e-3 | 1.40e-1 ± 1.9e-3 | 6.93e-2 ± 1.2e-3 | 2.11e-2 ± 6.9e-4 | **1.64e-2 ± 6.6e-4** |
| Birds 200 25-shot | MiX/L/16 | 1.12e-1 ± 2.0e-3 | 1.12e-1 ± 2.0e-3 | 1.12e-1 ± 2.0e-3 | 5.44e-2 ± 1.8e-3 | **2.08e-2 ± 1.1e-3** |
| Birds 200 25-shot | ViT/B/16 | 9.02e-2 ± 1.6e-3 | 9.02e-2 ± 1.6e-3 | 3.75e-2 ± 1.0e-3 | **1.51e-2 ± 5.7e-4** | 1.62e-2 ± 6.1e-4 |
| Birds 200 25-shot | ViT/S/16 | 5.06e-2 ± 1.4e-3 | 5.06e-2 ± 1.4e-3 | 4.96e-2 ± 1.4e-3 | 4.02e-2 ± 1.2e-3 | **1.03e-2 ± 6.6e-4** |
| Birds 200 5-shot | BiT/101/3 | 8.17e-2 ± 2.0e-3 | 8.17e-2 ± 2.0e-3 | 8.17e-2 ± 2.0e-3 | 3.38e-2 ± 1.3e-3 | **1.81e-2 ± 8.2e-4** |
| Birds 200 5-shot | BiT/50/1 | 5.44e-2 ± 5.6e-4 | 5.44e-2 ± 5.6e-4 | 5.44e-2 ± 5.6e-4 | 2.59e-2 ± 5.4e-4 | **1.34e-2 ± 3.7e-4** |
| Birds 200 5-shot | MiX/B/16 | 8.27e-2 ± 1.0e-3 | 8.27e-2 ± 1.0e-3 | 5.49e-2 ± 7.8e-4 | 2.14e-2 ± 5.3e-4 | **1.39e-2 ± 4.1e-4** |
| Birds 200 5-shot | MiX/L/16 | 5.68e-2 ± 1.4e-3 | 5.68e-2 ± 1.4e-3 | 5.68e-2 ± 1.4e-3 | 3.20e-2 ± 9.7e-4 | **1.85e-2 ± 6.4e-4** |
| Birds 200 5-shot | ViT/B/16 | 3.40e-2 ± 8.9e-4 | 3.40e-2 ± 8.9e-4 | 3.40e-2 ± 8.9e-4 | 1.65e-2 ± 6.7e-4 | **1.36e-2 ± 5.8e-4** |
| Birds 200 5-shot | ViT/S/16 | 2.75e-2 ± 7.9e-4 | 2.75e-2 ± 7.9e-4 | 2.75e-2 ± 7.9e-4 | 1.20e-2 ± 5.2e-4 | **7.39e-3 ± 4.5e-4** |
| CIFAR-100 10-shot | BiT/101/3 | 8.57e-2 ± 3.8e-3 | 8.57e-2 ± 3.8e-3 | 8.25e-2 ± 3.7e-3 | 4.77e-2 ± 3.0e-3 | **2.58e-2 ± 2.3e-3** |
| CIFAR-100 10-shot | BiT/50/1 | 7.44e-2 ± 1.5e-3 | 1.24e-2 ± 5.8e-4 | 2.08e-2 ± 7.2e-4 | **1.24e-2 ± 5.8e-4** | 1.83e-2 ± 8.3e-4 |
| CIFAR-100 10-shot | MiX/B/16 | 8.77e-2 ± 1.9e-3 | 8.77e-2 ± 1.9e-3 | 2.71e-2 ± 1.2e-3 | **2.37e-2 ± 9.9e-4** | 2.44e-2 ± 9.5e-4 |
| CIFAR-100 10-shot | MiX/L/16 | 1.05e-1 ± 3.1e-3 | 1.05e-1 ± 3.1e-3 | 4.85e-2 ± 2.6e-3 | 4.97e-2 ± 1.6e-3 | **4.75e-2 ± 2.6e-3** |
| CIFAR-100 10-shot | ViT/B/16 | 8.98e-2 ± 2.0e-3 | 8.98e-2 ± 2.0e-3 | 8.98e-2 ± 2.0e-3 | 4.98e-2 ± 1.7e-3 | **3.71e-2 ± 1.4e-3** |
| CIFAR-100 10-shot | ViT/S/16 | 6.84e-2 ± 1.1e-3 | **2.11e-2 ± 6.6e-4** | 3.35e-2 ± 8.6e-4 | 2.54e-2 ± 7.5e-4 | 2.57e-2 ± 7.5e-4 |
| CIFAR-100 25-shot | BiT/101/3 | 8.77e-2 ± 5.6e-3 | 8.77e-2 ± 5.6e-3 | 4.44e-2 ± 3.5e-3 | 3.40e-2 ± 2.7e-3 | **2.88e-2 ± 3.0e-3** |
| CIFAR-100 25-shot | BiT/50/1 | 7.31e-2 ± 2.0e-3 | 2.35e-2 ± 1.5e-3 | 3.65e-2 ± 1.8e-3 | 2.35e-2 ± 1.5e-3 | **1.89e-2 ± 1.1e-3** |
| CIFAR-100 25-shot | MiX/B/16 | 1.08e-1 ± 2.3e-3 | 4.75e-2 ± 1.6e-3 | **2.10e-2 ± 9.4e-4** | 2.24e-2 ± 9.9e-4 | 2.67e-2 ± 1.1e-3 |
| CIFAR-100 25-shot | MiX/L/16 | 9.79e-2 ± 2.2e-3 | 9.79e-2 ± 2.2e-3 | 3.67e-2 ± 1.7e-3 | **2.98e-2 ± 1.4e-3** | 3.45e-2 ± 1.6e-3 |
| CIFAR-100 25-shot | ViT/B/16 | 1.07e-1 ± 1.9e-3 | 1.07e-1 ± 1.9e-3 | 6.54e-2 ± 1.6e-3 | 4.80e-2 ± 1.4e-3 | **3.02e-2 ± 4.5e-3** |
| CIFAR-100 25-shot | ViT/S/16 | 8.03e-2 ± 1.2e-3 | 2.19e-2 ± 7.4e-4 | 3.13e-2 ± 8.4e-4 | 2.27e-2 ± 7.1e-4 | **2.14e-2 ± 6.9e-4** |
| CIFAR-100 5-shot | BiT/101/3 | 5.94e-2 ± 3.2e-3 | 5.94e-2 ± 3.2e-3 | 5.94e-2 ± 3.2e-3 | **3.30e-2 ± 2.4e-3** | 3.78e-2 ± 2.6e-3 |
| CIFAR-100 5-shot | BiT/50/1 | 4.87e-2 ± 1.3e-3 | 4.87e-2 ± 1.3e-3 | 1.69e-2 ± 8.8e-4 | 1.87e-2 ± 8.9e-4 | **1.45e-2 ± 8.7e-4** |
| CIFAR-100 5-shot | MiX/B/16 | 7.07e-2 ± 1.2e-3 | 7.07e-2 ± 1.2e-3 | 2.78e-2 ± 8.4e-4 | 1.76e-2 ± 6.6e-4 | **1.70e-2 ± 6.3e-4** |
| CIFAR-100 5-shot | MiX/L/16 | 7.06e-2 ± 1.6e-3 | 7.06e-2 ± 1.6e-3 | 4.17e-2 ± 1.4e-3 | 3.32e-2 ± 1.2e-3 | **2.77e-2 ± 1.0e-3** |
| CIFAR-100 5-shot | ViT/B/16 | 6.27e-2 ± 1.6e-3 | 6.27e-2 ± 1.6e-3 | 6.27e-2 ± 1.6e-3 | 4.30e-2 ± 1.3e-3 | **2.82e-2 ± 1.0e-3** |
| CIFAR-100 5-shot | ViT/S/16 | 6.93e-2 ± 1.2e-3 | **2.84e-2 ± 8.2e-4** | 3.88e-2 ± 8.0e-4 | 3.16e-2 ± 7.5e-4 | 3.50e-2 ± 9.2e-3 |
| Caltech101 10-shot | BiT/101/3 | 3.07e-1 ± 2.0e-2 | 3.07e-1 ± 2.0e-2 | 1.51e-1 ± 1.3e-2 | 1.00e-1 ± 1.1e-2 | **4.75e-2 ± 8.1e-3** |
| Caltech101 10-shot | BiT/50/1 | 3.29e-1 ± 1.6e-2 | 7.68e-2 ± 5.0e-3 | 1.13e-1 ± 6.0e-3 | 6.01e-2 ± 4.4e-3 | **1.77e-2 ± 2.5e-3** |
| Caltech101 10-shot | MiX/B/16 | **1.35e-1 ± 1.4e-2** | 1.35e-1 ± 1.4e-2 | 1.35e-1 ± 1.4e-2 | 1.92e-1 ± 1.6e-2 | 2.04e-1 ± 9.7e-3 |
| Caltech101 10-shot | MiX/L/16 | 1.25e-1 ± 1.3e-2 | 1.25e-1 ± 1.3e-2 | **1.25e-1 ± 1.3e-2** | 1.30e-1 ± 1.2e-2 | 2.13e-1 ± 1.5e-2 |
| Caltech101 10-shot | ViT/B/16 | 7.76e-2 ± 4.3e-3 | 7.76e-2 ± 4.3e-3 | **3.11e-2 ± 3.0e-3** | 5.75e-2 ± 4.4e-3 | 4.02e-2 ± 3.9e-3 |
| Caltech101 10-shot | ViT/S/16 | 1.95e-1 ± 6.0e-3 | 3.41e-2 ± 2.9e-3 | **2.40e-2 ± 2.0e-3** | 3.41e-2 ± 2.9e-3 | 2.40e-2 ± 2.0e-3 |
| Caltech101 25-shot | BiT/101/3 | 1.15e-1 ± 6.5e-3 | 1.15e-1 ± 6.5e-3 | 1.15e-1 ± 6.5e-3 | 1.15e-1 ± 6.5e-3 | **9.86e-2 ± 8.0e-3** |
| Caltech101 25-shot | BiT/50/1 | 3.60e-1 ± 1.9e-2 | 8.80e-2 ± 5.5e-3 | 1.43e-1 ± 7.6e-3 | 4.76e-2 ± 3.6e-3 | **1.55e-2 ± 1.6e-3** |
| Caltech101 25-shot | MiX/B/16 | **8.28e-2 ± 1.2e-2** | 8.28e-2 ± 1.2e-2 | 8.28e-2 ± 1.2e-2 | 1.65e-1 ± 1.7e-2 | 1.93e-1 ± 1.3e-2 |
| Caltech101 25-shot | MiX/L/16 | 9.66e-2 ± 1.0e-2 | 9.66e-2 ± 1.0e-2 | 9.66e-2 ± 1.0e-2 | **9.66e-2 ± 1.0e-2** | 1.49e-1 ± 1.3e-2 |
| Caltech101 25-shot | ViT/B/16 | 1.03e-1 ± 5.6e-3 | **3.33e-2 ± 2.5e-3** | 4.46e-2 ± 3.6e-3 | 3.33e-2 ± 2.5e-3 | 3.95e-2 ± 5.4e-3 |
| Caltech101 25-shot | ViT/S/16 | 1.77e-1 ± 5.4e-3 | 3.79e-2 ± 3.1e-3 | **2.80e-2 ± 1.8e-3** | 3.79e-2 ± 3.1e-3 | 3.29e-2 ± 2.1e-3 |
| Caltech101 5-shot | BiT/101/3 | 2.12e-1 ± 1.2e-2 | 2.12e-1 ± 1.2e-2 | 2.12e-1 ± 1.2e-2 | 1.65e-1 ± 9.4e-3 | **1.87e-2 ± 4.3e-3** |
| Caltech101 5-shot | BiT/50/1 | 2.34e-1 ± 6.1e-3 | 4.13e-2 ± 2.1e-3 | **1.61e-2 ± 1.3e-3** | 4.69e-2 ± 2.1e-3 | 4.10e-2 ± 2.1e-3 |
| Caltech101 5-shot | MiX/B/16 | 2.43e-1 ± 1.2e-2 | 2.43e-1 ± 1.2e-2 | 2.35e-1 ± 1.1e-2 | 7.28e-2 ± 4.3e-3 | **1.92e-2 ± 1.9e-3** |
| Caltech101 5-shot | MiX/L/16 | 1.38e-1 ± 9.7e-3 | 1.38e-1 ± 9.7e-3 | 1.38e-1 ± 9.7e-3 | **1.37e-1 ± 9.9e-3** | 1.63e-1 ± 1.1e-2 |
| Caltech101 5-shot | ViT/B/16 | 1.10e-1 ± 6.3e-3 | 1.10e-1 ± 6.3e-3 | 6.02e-2 ± 4.7e-3 | 6.81e-2 ± 4.8e-3 | **3.87e-2 ± 3.4e-3** |
| Caltech101 5-shot | ViT/S/16 | 1.90e-1 ± 4.7e-3 | 3.82e-2 ± 2.6e-3 | 5.04e-2 ± 2.9e-3 | 3.82e-2 ± 2.6e-3 | **2.78e-2 ± 1.8e-3** |
| ImageNet 10-shot | BiT/101/3 | 1.27e-1 ± 2.0e-3 | 1.27e-1 ± 2.0e-3 | 7.36e-2 ± 1.1e-3 | 3.06e-2 ± 7.0e-4 | **6.65e-3 ± 3.8e-4** |
| ImageNet 10-shot | BiT/50/1 | 9.54e-2 ± 7.2e-4 | 9.54e-2 ± 7.2e-4 | 5.75e-3 ± 2.0e-4 | 1.86e-2 ± 2.8e-4 | **3.84e-3 ± 1.5e-4** |
| ImageNet 10-shot | MiX/B/16 | 9.34e-2 ± 7.9e-4 | 9.34e-2 ± 7.9e-4 | 3.37e-2 ± 2.9e-4 | 2.32e-2 ± 3.0e-4 | **4.22e-3 ± 1.5e-4** |
| ImageNet 10-shot | MiX/L/16 | 9.83e-2 ± 1.3e-3 | 9.83e-2 ± 1.3e-3 | 9.83e-2 ± 1.3e-3 | **4.01e-3 ± 1.9e-4** | 4.33e-3 ± 1.8e-4 |
| ImageNet 10-shot | ViT/B/16 | 4.62e-2 ± 7.1e-4 | 4.62e-2 ± 7.1e-4 | 4.62e-2 ± 7.1e-4 | 1.44e-2 ± 3.0e-4 | **5.70e-3 ± 2.0e-4** |
| ImageNet 10-shot | ViT/S/16 | 4.74e-2 ± 5.6e-4 | 4.74e-2 ± 5.6e-4 | 1.66e-2 ± 2.5e-4 | 7.18e-3 ± 2.0e-4 | **3.71e-3 ± 1.4e-4** |
| ImageNet 25-shot | BiT/101/3 | 1.42e-1 ± 2.3e-3 | 1.42e-1 ± 2.3e-3 | 6.67e-2 ± 9.1e-4 | 3.31e-2 ± 8.7e-4 | **4.76e-3 ± 2.8e-4** |
| ImageNet 25-shot | BiT/50/1 | 1.17e-1 ± 9.2e-4 | 1.17e-1 ± 9.2e-4 | **4.06e-3 ± 1.7e-4** | 1.84e-2 ± 2.6e-4 | 4.67e-3 ± 1.6e-4 |
| ImageNet 25-shot | MiX/B/16 | 9.59e-2 ± 9.3e-4 | 9.59e-2 ± 9.3e-4 | 5.39e-2 ± 4.9e-4 | 2.04e-2 ± 3.1e-4 | **4.17e-3 ± 1.7e-4** |
| ImageNet 25-shot | MiX/L/16 | 1.03e-1 ± 1.3e-3 | 1.03e-1 ± 1.3e-3 | 1.03e-1 ± 1.3e-3 | **6.33e-3 ± 2.2e-4** | 7.60e-3 ± 2.6e-4 |
| ImageNet 25-shot | ViT/B/16 | 5.17e-2 ± 8.8e-4 | 5.17e-2 ± 8.8e-4 | 5.17e-2 ± 8.8e-4 | 1.52e-2 ± 3.8e-4 | **4.96e-3 ± 2.0e-4** |
| ImageNet 25-shot | ViT/S/16 | 5.52e-2 ± 4.4e-4 | 4.12e-2 ± 3.4e-4 | 9.65e-3 ± 2.3e-4 | 7.78e-3 ± 2.1e-4 | **6.11e-3 ± 2.4e-4** |
| ImageNet 5-shot | BiT/101/3 | 9.24e-2 ± 1.4e-3 | 9.24e-2 ± 1.4e-3 | 9.24e-2 ± 1.4e-3 | 2.09e-2 ± 7.9e-4 | **8.05e-3 ± 5.0e-4** |
| ImageNet 5-shot | BiT/50/1 | 8.95e-2 ± 6.7e-4 | 8.95e-2 ± 6.7e-4 | 1.53e-2 ± 2.2e-4 | 1.11e-2 ± 2.3e-4 | **7.94e-3 ± 2.1e-4** |
| ImageNet 5-shot | MiX/B/16 | 9.09e-2 ± 7.2e-4 | 9.09e-2 ± 7.2e-4 | 3.01e-2 ± 2.8e-4 | 1.95e-2 ± 2.7e-4 | **6.49e-3 ± 2.2e-4** |
| ImageNet 5-shot | MiX/L/16 | 7.99e-2 ± 9.7e-4 | 7.99e-2 ± 9.7e-4 | 7.99e-2 ± 9.7e-4 | 9.92e-3 ± 4.5e-4 | **5.68e-3 ± 2.4e-4** |
| ImageNet 5-shot | ViT/B/16 | 4.11e-2 ± 6.3e-4 | 4.11e-2 ± 6.3e-4 | 4.11e-2 ± 6.3e-4 | 1.55e-2 ± 2.8e-4 | **1.29e-2 ± 2.7e-4** |
| ImageNet 5-shot | ViT/S/16 | 4.20e-2 ± 4.1e-4 | 4.20e-2 ± 4.1e-4 | 2.40e-2 ± 2.6e-4 | 8.02e-3 ± 1.9e-4 | **4.72e-3 ± 1.6e-4** |

Table 3: Extrapolation Results on scaling behavior of Downstream Vision Tasks. See Section 5.1 for more details. Numbers for M1, M2, M3, and M4 obtained via correspondence with authors of Alabdulmohsin et al. (2022).

## 5.2 LANGUAGE

Using the scaling laws benchmark of Alabdulmohsin et al. (2022), we evaluate how well various functional forms extrapolate performance on language tasks as the (pre-)training dataset size increases. In this large-scale language subset of the benchmark, the tasks that are evaluated are error rates on each of the various downstream tasks from the BIG-Bench (BB) (Srivastava et al., 2022) benchmark and upstream test cross-entropy of various models trained to do language modeling (LM) and neural machine translation (NMT). All LM and BB tasks use a decoder-only language model. As can be seen in Tables 2 and 4, BNSL yields extrapolations with the lowest RMSLE (Root Mean Squared Logarithmic Error) for 75% of tasks of any of the functional forms, while the next best functional form performs the best on only 10% of the tasks.

To view all plots of the BNSL on each of these tasks, see Figures 27, 28, 29 in Appendix A.26. To view plots of M1, M2, M3, and M4 on these tasks, see Figure 8 of Alabdulmohsin et al. (2022).

In Section A.13, we additionally show that BNSL yields accurate extrapolations of performance on large-scale downstream language tasks when number of model parameters is on the x-axis.

In Section A.14, we show that BNSL accurately models and extrapolates the scaling behavior of sparse models (i.e. sparse, pruned models and sparsely gated mixture-of-expert models).

In Section A.25, we additionally show that BNSL yields accurate extrapolations of performance on large-scale downstream audio (speech recognition) tasks.

In Section A.16, we show BNSL accurately models and extrapolates the scaling behavior with fine-tuning dataset size on the x-axis and the scaling behavior of computer programming / coding.

In Section A.23, BNSL accurately extrapolates the scaling behavior of math word problems.

In Section A.22, BNSL accurately extrapolates the scaling behavior of tasks involving molecules.

In Section A.21, BNSL accurately extrapolates the scaling behavior of OOD detection.

In Section A.17, we additionally show BNSL accurately models and extrapolates the scaling behavior of uncertainty estimation / calibration.

| Domain | Task | Model | M1 ↓ | M2 ↓ | M3 ↓ | M4 ↓ | BNSL ↓ |
|---|---|---|---|---|---|---|---|
| BB | date understanding, 1-shot | 2.62e+8 Param | 3.19e-2 ± 9.6e-4 | 3.19e-2 ± 9.6e-4 | 4.67e-3 ± 1.4e-4 | 3.19e-2 ± 9.6e-4 | **3.40e-3 ± 7.9e-5** |
| BB | date understanding, 2-shot | 2.62e+8 Param | 2.86e-2 ± 6.2e-4 | 2.86e-2 ± 6.2e-4 | 4.83e-3 ± 4.1e-4 | 2.86e-2 ± 6.2e-4 | **4.38e-3 ± 4.0e-4** |
| BB | linguistic mappings, 1-shot | 2.62e+8 Param | 1.66e-2 ± 5.5e-4 | 1.62e-2 ± 5.4e-4 | 1.66e-2 ± 5.5e-4 | 1.33e-2 ± 3.8e-4 | **1.13e-2 ± 2.2e-4** |
| BB | linguistic mappings, 2-shot | 2.62e+8 Param | 1.70e-2 ± 6.5e-4 | 1.70e-2 ± 6.5e-4 | 1.70e-2 ± 6.5e-4 | 1.06e-2 ± 5.1e-4 | **9.51e-3 ± 5.1e-4** |
| BB | mult data wrangling, 1-shot | 2.62e+8 Param | 1.07e-2 ± 1.0e-3 | 1.07e-2 ± 1.0e-3 | 1.07e-2 ± 1.0e-3 | 6.66e-3 ± 7.3e-4 | **6.39e-3 ± 4.6e-4** |
| BB | mult data wrangling, 2-shot | 2.62e+8 Param | 1.57e-2 ± 1.5e-3 | 1.57e-2 ± 1.5e-3 | 1.57e-2 ± 1.5e-3 | 5.79e-3 ± 7.0e-4 | **2.67e-3 ± 2.7e-4** |
| BB | qa wikidata, 1-shot | 2.62e+8 Param | **4.27e-3 ± 8.9e-4** | 4.32e-3 ± 8.2e-4 | 4.27e-3 ± 8.9e-4 | 4.32e-3 ± 8.2e-4 | 4.68e-3 ± 7.3e-4 |
| BB | qa wikidata, 2-shot | 2.62e+8 Param | **4.39e-3 ± 7.0e-4** | 4.66e-3 ± 6.4e-4 | 4.39e-3 ± 7.0e-4 | 9.02e-3 ± 6.9e-4 | 8.05e-3 ± 7.3e-4 |
| BB | unit conversion, 1-shot | 2.62e+8 Param | 8.30e-3 ± 4.4e-4 | 8.30e-3 ± 4.4e-4 | **1.48e-3 ± 2.7e-4** | 4.79e-3 ± 3.4e-4 | 1.07e-2 ± 2.5e-4 |
| BB | unit conversion, 2-shot | 2.62e+8 Param | 1.07e-2 ± 4.4e-4 | 1.07e-2 ± 4.4e-4 | 7.50e-3 ± 5.5e-4 | 7.55e-3 ± 5.1e-4 | **7.02e-3 ± 3.9e-4** |
| LM | upstream test cross-entropy | 1.07e+9 Param | 1.71e-2 ± 6.0e-4 | 1.66e-3 ± 5.1e-5 | 4.50e-3 ± 5.9e-5 | 1.28e-3 ± 3.9e-5 | **9.71e-4 ± 3.2e-5** |
| LM | upstream test cross-entropy | 4.53e+8 Param | 1.65e-2 ± 6.6e-4 | 7.41e-4 ± 9.8e-5 | 6.58e-4 ± 6.6e-5 | 7.41e-4 ± 9.8e-5 | **5.86e-4 ± 7.7e-5** |
| LM | upstream test cross-entropy | 2.62e+8 Param | 1.55e-2 ± 7.2e-4 | 9.20e-4 ± 9.7e-5 | 3.97e-3 ± 1.3e-4 | 9.20e-4 ± 9.7e-5 | **7.90e-4 ± 5.1e-5** |
| LM | upstream test cross-entropy | 1.34e+8 Param | 1.43e-2 ± 4.8e-4 | 1.46e-3 ± 6.8e-5 | **6.46e-4 ± 5.1e-5** | 1.46e-3 ± 6.8e-5 | 9.01e-4 ± 5.5e-5 |
| LM | upstream test cross-entropy | 1.68e+7 Param | 6.37e-3 ± 9.4e-5 | **3.03e-4 ± 1.2e-5** | 1.56e-3 ± 3.5e-5 | 3.03e-4 ± 1.2e-5 | 4.34e-4 ± 1.8e-5 |
| NMT | upstream test cross-entropy | 28 Enc, 6 Dec | 1.71e-1 ± 0 | 5.64e-2 ± 0 | 3.37e-2 ± 0 | 1.81e-2 ± 0 | **1.69e-2 ± 0** |
| NMT | upstream test cross-entropy | 6 Enc, 28 Dec | 2.34e-1 ± 0 | 5.27e-2 ± 0 | 1.65e-2 ± 0 | 4.44e-2 ± 0 | **1.56e-2 ± 0** |
| NMT | upstream test cross-entropy | 6 Enc, 6 Dec | 2.62e-1 ± 0 | 3.84e-2 ± 0 | 8.92e-2 ± 0 | 2.05e-2 ± 0 | **1.37e-3 ± 0** |
| NMT | upstream test cross-entropy | Dec-only, LM | 2.52e-1 ± 0 | 1.03e-2 ± 0 | 3.28e-2 ± 0 | 8.43e-3 ± 0 | **7.33e-3 ± 0** |
| NMT | upstream test cross-entropy | Transformer-Enc, LSTM-Dec | 1.90e-1 ± 0 | 1.26e-2 ± 0 | 6.32e-2 ± 0 | 1.26e-2 ± 0 | **8.30e-3 ± 0** |

Table 4: Extrapolation Results on scaling behavior of Language Tasks. See Section 5.2 for more details. Numbers for M1, M2, M3, and M4 were obtained via correspondence with authors of Alabdulmohsin et al. (2022). BB stands for BIG-Bench (Srivastava et al., 2022). NMT stands for Neural Machine Translation. LM stands for Language Modeling.

## 5.3 REINFORCEMENT LEARNING

We show that BNSL accurately models and extrapolates the scaling behaviors of various multi-agent and single-agent reinforcement learning algorithms trained in various environments. In the top left plot and top right plot and bottom left plot of Figure 2, BNSL accurately models and extrapolates the scaling behavior of the AlphaZero algorithm trained to play the game Connect Four from Figure 4 and Figure 5 and Figure 3 respectively of Neumann & Gros (2022); the x-axes respectively are compute (FLOPs) used for training, training dataset size (states), and number of model parameters. In Figure 2 bottom left and bottom right respectively, BNSL accurately models and extrapolates the scaling behavior of the Phasic Policy Gradient (PPG) algorithm (Cobbe et al., 2021b) trained to play the Procgen (Cobbe et al., 2020) game called StarPilot and the scaling behavior of the Proximal Policy Optimization (PPO) algorithm (Schulman et al., 2017) trained to play the Procgen (Cobbe et al., 2020) game called Heist.

In Section A.18, we find BNSL accurately extrapolates the scaling behavior of a pretrained language model finetuned (i.e. aligned) via Reinforcement Learning from Human Feedback (RLHF) to be helpful from Figure 1 of Bai et al. (2022).

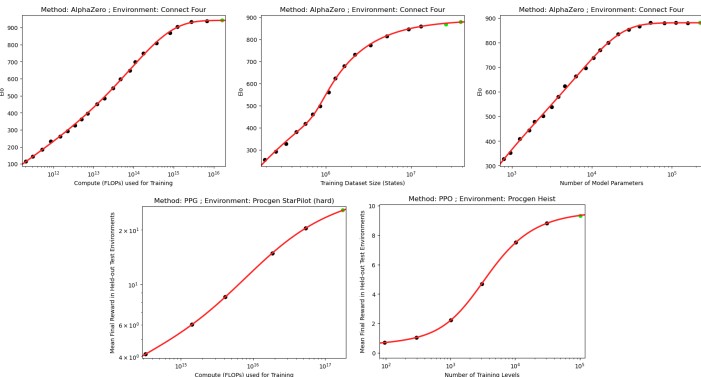

Figure 2: Extrapolation of BNSL on Reinforcement Learning Scaling Experimental Data. Experimental data of the top left plot and top middle plot and top right plot is from Figure 4 and Figure 5 and Figure 3 respectively of Neumann & Gros (2022). Experimental Data of the bottom left plot is from Figure 1 left of Hilton et al. (2023). Experimental Data of the bottom right plot is from Figure 2 of Cobbe et al. (2020). Top left and bottom left plot is the compute-optimal Pareto frontier. See Section 5.3 for more details.

## 5.4 NON-MONOTONIC SCALING

We show that BNSL accurately models and extrapolates non-monotonic scaling behaviors that are exhibited by Transformers (Vaswani et al. (2017)) in double descent (Nakkiran et al., 2021) in Figure 3. Various other functional forms are mathematically incapable of expressing non-monotonic behaviors (as shown in Section 4).

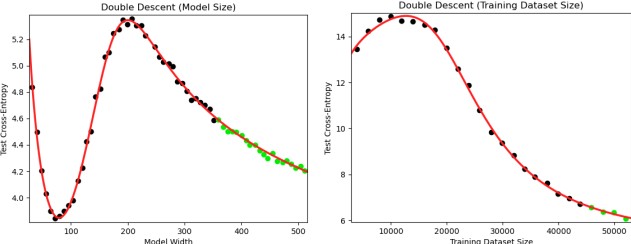

Figure 3: Extrapolation of BNSL on Double Descent. Both plots are of transformers trained to do neural machine translation via minimizing cross-entropy. Experimental data of left figure is obtained from Figure 8 top of Nakkiran et al. (2021); "Model Width" on the x-axis refers to embedding dimension $d_{model}$ of the transformer; note that model width is linearly proportional to number of model parameters, so number of model parameters on the x-axis would yield same results. Experimental data of the right figure is obtained from Figure 11b of Nakkiran et al. (2021). The plot on the left contains **two breaks** of a BNSL fit to the black points. See Section 5.4 for more details.

## 5.5 INFLECTION POINTS

We show that BNSL is capable of modeling and extrapolating the scaling behavior of tasks that have an inflection point on a linear-linear plot such as the task of arithmetic (4-digit addition). Here we model and extrapolate the scaling behavior of a transformer model (Vaswani et al. (2017)) with respect to the training dataset size on the 4-digit addition task. Various other functional forms are mathematically incapable of expressing inflection points on a linear-linear plot (as shown in Section 4) and as a result, are mathematically incapable of expressing and modeling inflection points (on a linear-linear plot) that are present in the scaling behavior of 4-digit addition. In Figure 4 left, we show that BNSL expresses and accurately models the inflection point present in the scaling behavior of 4-digit addition and as a result accurately extrapolates the scaling behavior of 4 digit addition. For further details about the hyperparameters please refer to the Appendix Section A.5.

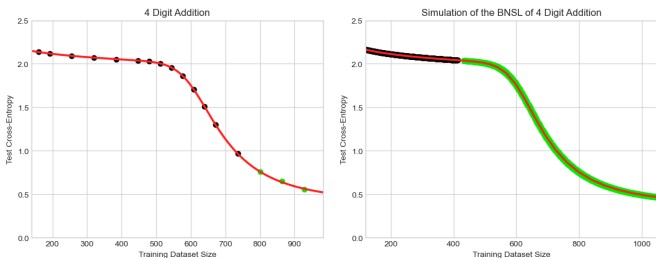

Figure 4: Extrapolation of BNSL on 4 Digit Addition. Note these plots are linear-linear. Each point in left plot is mean of greater than 1000 seeds at that dataset size. In left plot, each point is gathered from a model trained to do task of 4 digit addition. In right plot, each point is gathered from a noiseless simulation of the BNSL of the task of 4 digit addition. See Sections 5.5, A.5, 6, for more details.

## 6    THE LIMIT OF THE PREDICTABILITY OF SCALING BEHAVIOR

We use BNSL to glean insights about the limit of the predictability of scaling behavior. Recent papers (Ganguli et al., 2022; Wei et al., 2022a) have advertised many tasks as having "unpredictable" "emergent" "phase transition/change" scaling behavior, the most famous of which is the task of arithmetic. In the previous section and in Figure 4 left, we successfully predicted (i.e. extrapolated) the scaling behavior of 4-digit addition (arithmetic). However, we are only able to accurately extrapolate the scaling behavior if given some points from training runs with a training dataset size of at least 720, and the break in which the scaling behavior of 4-digit addition transitions from one power law to another steeper power-law happens at around training dataset size of 415.

Ideally, one would like to be able to extrapolate the entire scaling behavior by fitting only points from before the break. In Figure 4 right, we use a noiseless simulation of the BNSL of 4-digit addition to show what would happen if one had infinitely many training runs / seeds to average out all the noisy deviation between runs such that one could recover (i.e. learn via a curve-fitting library such as SciPy (Virtanen et al., 2020)) the learned constants of the BNSL as well as possible. When using this noiseless simulation, we find that we are only able to accurately extrapolate the scaling behavior if given some points from training runs with a training dataset size of at least 415, which is very close to the break.

This has a few implications:

1) When the scaling behavior exhibits greater than 0 breaks that are sufficiently sharp, there is a limit as to how small the maximum (along the x-axis) of the points used for fitting can be if one wants to perfectly extrapolate the scaling behavior, even if one has infinitely many seeds / training runs.

2) If an additional break of sufficient sharpness happens at a scale that is sufficiently larger than the maximum (along the x-axis) of the points used for fitting, there does not (currently) exist a way to extrapolate the scaling behavior after that additional break.

3) If a break of sufficient sharpness happens at a scale sufficiently smaller than the maximum (along the x-axis) of the points used for fitting, points smaller (along the x-axis) than that break are often useless for improving extrapolation.

## 7    CONCLUSIONS

We have presented a smoothly broken power law functional form that accurately models and extrapolates the scaling behaviors of artificial neural networks for various architectures and for each of various tasks from a very large and diverse set of upstream and downstream tasks. This set includes large-scale vision, language, audio, video, diffusion, generative modeling, multimodal learning, contrastive learning, AI alignment, robotics, out-of-distribution generalization, continual learning, uncertainty estimation / calibration, out-of-distribution detection, adversarial robustness, molecules, computer programming/coding, math word problems, arithmetic, unsupervised/self-supervised learning, and reinforcement learning (single agent and multi-agent). When compared to other functional forms for neural scaling behavior, this functional form yields extrapolations of scaling behavior that are considerably more accurate on this set. Additionally, this functional form accurately models and extrapolates scaling behavior that other functional forms are incapable of expressing such as the non-monotonic transitions present in the scaling behavior of phenomena such as double descent and the delayed, sharp inflection points present in the scaling behavior of tasks such as arithmetic. Lastly, we used this functional form to glean insights about the limit of the predictability of scaling behavior. See arXiv for longer version.

ETHICS STATEMENT

We place relatively high probability on the claim that variants of smoothly broken power laws perhaps are the "true" functional form of the scaling behavior of all(?) things that involve artificial neural networks. Due to the fact that BNSL is a variant of smoothly broken power laws, an ethical concern one might have about our work is that revealing BNSL might differentially (Hendrycks & Mazeika, 2022) improve A(G)I capabilities progress relative to A(G)I safety/alignment progress. A counter-argument is that BNSL will also allow the A(G)I safety/alignment field to extrapolate the scaling behaviors of its methods for aligning A(G)I systems and as a result will also accelerate alignment/safety progress. Existing scaling laws besides BNSL struggle especially to model downstream performance, e.g. on safety-relevant evaluations (especially evaluations (such as interpretability and controllability) that might exhibit non-monotonic scaling behavior in the larger scale systems of the future); we believe our work could differentially help in forecasting emergence of novel capabilities (such as reasoning (Wei et al., 2022b)) or behaviors (such as deception or dishonesty (Evans et al., 2021; Lin et al., 2021)), and thus help avoid unpleasant surprises.

A potential limitation of the current approach is the need to collect enough samples of the system's performance (i.e. the (x,y) points required for estimating the scaling laws parameters). A small number of samples sometimes may not be sufficient to accurately fit and extrapolate the BNSL functional form, and obtaining a large number of such samples can sometimes be costly. This has the ethical implication that entities with more compute to gather more points maybe will have considerably more accurate extrapolations of scaling behavior than entities with less compute. As a result, entities with less compute (e.g. academia) maybe will have less foresight than entities with more compute (e.g. Big Tech), which could maybe exacerbate the gap between entities with more compute (e.g. Big Tech) and entities with less compute (e.g. academia).

ACKNOWLEDGMENTS

We are thankful for useful feedback and assistance from Kartik Ahuja, Ibrahim Alabdulmohsin, Ankesh Anand, Jacob Buckman, Guillaume Dumas, Leo Gao, Andy Jones, Behnam Neyshabur, Gabriel Prato, Stephen Roller, Michael Trazzi, Tony Wu and others.

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

# A    APPENDIX

## A.1    ANALYSIS AND EXPLANATION OF WHY BNSL IS SMOOTHLY CONNECTED PIECEWISE (APPROXIMATELY) LINEAR FUNCTION ON A LOG-LOG PLOT

Analysing Equation 1 reveals why BNSL is smoothly connected piecewise (approximately) linear function on a log-log plot. Considering $y$ as a function of $z := \log(x)$, applying logarithms to both sides and setting $a = 0$ yields:

$$\log(y) = \log(b) - c_0 z - \sum_{i=1}^{n} c_i f_i \log\left(1 + \left(\frac{\exp(z)}{d_i}\right)^{1/f_i}\right). \tag{3}$$

We can now see the terms in the sum resemble the well-known softplus function: $\text{softplus}(x) := \log(1 + exp(x))$, which smoothly interpolates between the constant 0 function and the identity. By plotting one such term for different values of $c_i, d_i, f_i$, it is easy to confirm that they influence the shape of the curve as described in Section 2.

## A.2    DECOMPOSITION OF BROKEN NEURAL SCALING LAW INTO POWER LAW SEGMENTS THAT IT IS COMPOSED OF

We now show a way to decompose a BNSL (Equation 1) with 3 breaks into the power law segments that it is composed of. This decomposition is what we used to produce segments 1-4 overlaid in Figure 1 and is usable when values of $f$ in Equation 1 are not too large. This decomposition pattern is straight-forward to extend to $n$ breaks.

$segment_1 = b * (x)^{-(c_0)}$

$segment_2 = b * (d_1)^{-(c_0)} * (x \ /d_1)^{-(c_1 + c_0)}$

$segment_3 = b * (d_1)^{-(c_0)} * (d_2/d_1)^{-(c_1 + c_0)} * (x \ /d_2)^{-(c_2 + c_1 + c_0)}$

$segment_4 = b * (d_1)^{-(c_0)} * (d_2/d_1)^{-(c_1 + c_0)} * (d_3/d_2)^{-(c_2 + c_1 + c_0)} * (x \ /d_3)^{-(c_3 + c_2 + c_1 + c_0)}$

## A.3    DEFINITION OF ROOT MEAN SQUARED LOG ERROR

$$Root\_Mean\_Squared\_Log\_Error = RMSLE = \sqrt{\left(\sum_{i=1}^{n}(log(y_i) - log(\hat{y}_i))^2\right)/n}$$

## A.4    DEFINITION OF ROOT STANDARD LOG ERROR

$$error = (log(y_i) - log(\hat{y}_i))^2)$$

$$\mu_{error} = \frac{1}{N}\sum_{i=1}^{N} error$$

$$\sigma_{error} = \sqrt{\frac{1}{N-1}\sum_{i=1}^{N}(error_i - \mu_{error})^2}$$

$$Root\_Standard\_Log\_Error = \sqrt{\mu_{error} + \frac{\sigma_{error}}{\sqrt{len(\hat{y})}}} - \sqrt{\mu_{error}}$$

## A.5    EXPERIMENTAL DETAILS OF SECTION 5.5

We perform an extensive set of experiments to model and extrapolate the scaling behavior for the 4-digit arithmetic addition task with respect to the training dataset size. Our code is based on the minGPT implementation (Karpathy, 2020). We set the batch size equal to the training dataset size. We do not use dropout or a learning rate decay here. Each experiment was run on a single V100 GPU and each run took less than 2 hours. For our experiments we train the transformer model using the following set of hyperparameters:

| | |
|---|---|
| $D_{model}$ | 128 |
| $D_{MLP}$ | 512 |
| Number of heads | 2 |
| Number of transformer blocks (i.e. layers) | 1 |
| Learning rate | 0.0001 |
| Weight Decay | 0.1 |
| Dropout Probability | 0.0 |
| Dataset sizes | 144-1008 |
| Vocab Size | 10 |

Table 5: Hyperparameters for 4-digit addition task

## A.6 Experimental details of fitting BNSL

We fit BNSL as follows: We first use scipy.optimize.brute to do a grid search of the values of the constants $(a, b, c_0, c_1...c_n, d_1...d_n, f_1...f_n)$ of BNSL that best minimize the mean squared log error (MSLE) between the real data and the output of BNSL. We then use the values obtained from the grid search as the initialization of the non-linear least squares algorithm of scipy.optimize.curve_fit. We then use the non-linear least squares algorithm of scipy.optimize.curve_fit to minimize the mean squared log error (MSLE) between the real data and the output of BNSL.

The version of MSLE we use for such optimization is the following numerically stable variant:

$$Numerically\_Stable\_MSLE = \sum_{i=1}^{n} ((log(y_i + 1) - log(\hat{y}_i + 1))^2)/n$$

## A.7 EXTRAPOLATION TO SCALES THAT ARE AN ORDER OF MAGNITUDE LARGER THAN THE MAXIMUM (ALONG THE X-AXIS) OF THE POINTS USED FOR FITTING

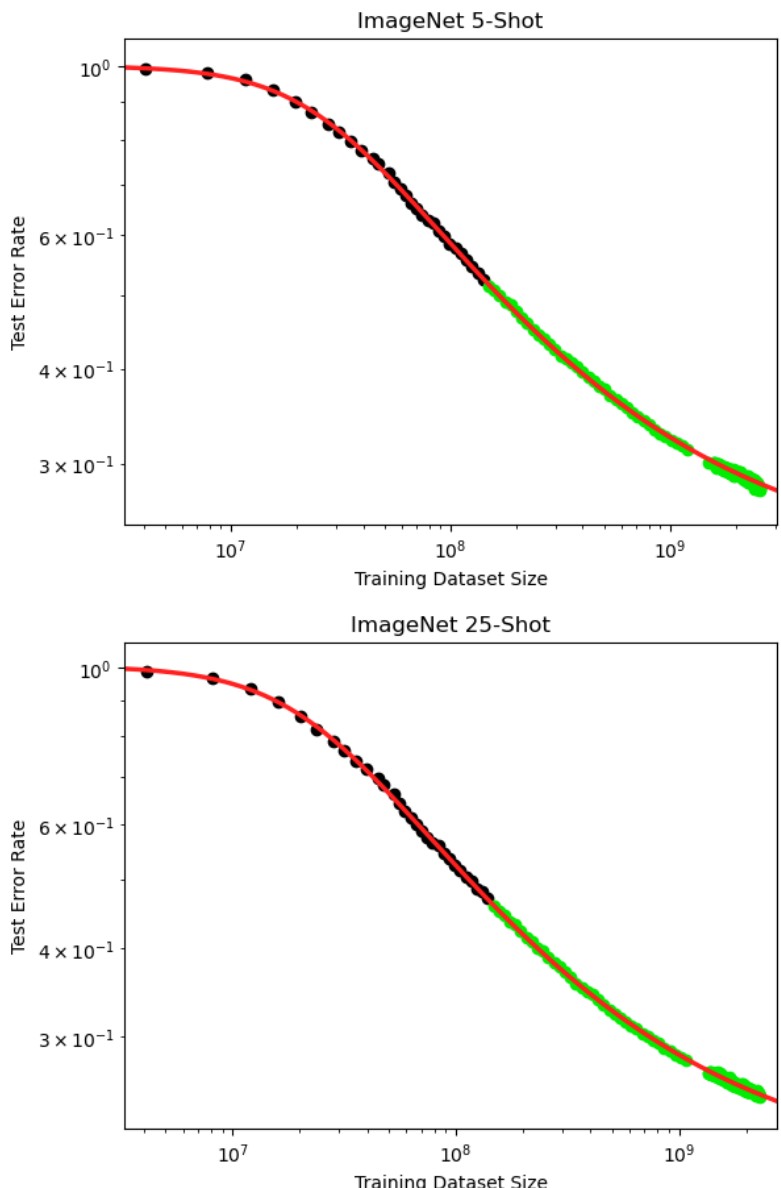

Figure 5: Extrapolation Results of BNSL to Scales that are an Order of Magnitude larger than the maximum (along the x-axis) of the points used for fitting. Experimental data of scaling behavior obtained from scaling laws benchmark of Alabdulmohsin et al. (2022). The upstream task is supervised pretraining of MLP mixers (MiX) (Tolstikhin et al., 2021) on subsets (i.e. the x-axis of plot) of JFT-300M (Sun et al., 2017). The Downstream Task is n-shot ImageNet classification (i.e. the y-axis of plot). See Section A.7 for more details.

In Figure 5, we show that BNSL accurately extrapolates to scales that are an order of magnitude larger than the maximum (along the x-axis) of the points used for fitting. The upstream task is supervised pretraining of MLP mixers (MiX) (Tolstikhin et al., 2021) on subsets (i.e. the x-axis of plot) of JFT-300M (Sun et al., 2017). The downstream task is n-shot ImageNet classification (i.e. the y-axis of plot). The experimental data of this scaling behavior is obtained from Alabdulmohsin et al. (2022).

### A.8 EXTRAPOLATION RESULTS FOR DOWNSTREAM VISION TASKS WHEN TRAINING RUNS ARE SCALED TO BE COMPUTE-OPTIMAL.

| Task | Model | M3 ↓ | BNSL ↓ |
|---|---|---|---|
| ImageNet 10-Shot | ViT | 1.91e-2 ± 6.48e-3 | **9.79e-3 ± 4.70e-3** |
| ImageNet Finetune | ViT | 1.14e-2 ± 2.42e-3 | **9.37e-3 ± 2.60e-3** |

Table 6: Extrapolation Results for Downstream Vision Tasks when training runs are scaled using the compute-optimal scaling (i.e. Pareto frontier) with respect to downstream performance. Experimental data obtained from Figure 2 of Zhai et al. (2021). See Section A.8 for more details.

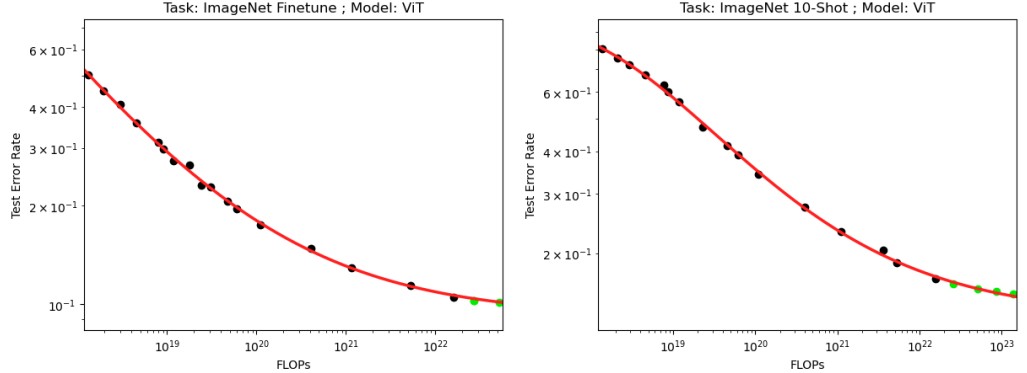

Figure 6: Extrapolation Results of BNSL for Downstream Vision Tasks when training runs are scaled to be compute-optimal. Experimental data obtained from Figure 2 of Zhai et al. (2021). See Section A.8 for more details.

In Figure 6 via fitting BNSL, we additionally obtain accurate extrapolations of scaling behavior of large-scale downstream vision tasks when compute (FLOPs) used for (pre-)training is on the x-axis and compute is scaled in the manner that is Pareto optimal with respect to the performance evaluation metric (downstream accuracy in this case). The experimental scaling data was obtained from Figure 2 of Zhai et al. (2021), and as a result in Table 6 we compare extrapolation of BNSL to the extrapolation of M3 (which was proposed in Zhai et al. (2021)); we find that BNSL that yields extrapolations of scaling behavior that are more accurate on these tasks.

## A.9 Extrapolation Results for Diffusion Generative Models of Images

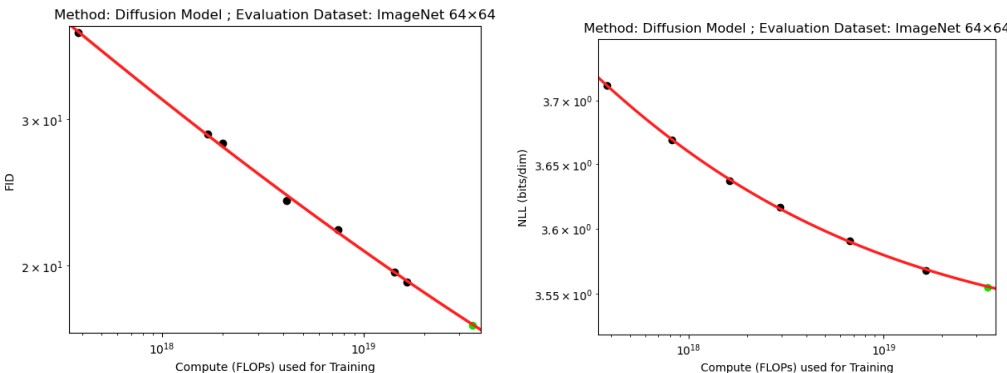

Figure 7: Extrapolation Results of BNSL for scaling behavior of Diffusion Generative Models of Images. Frechet Inception Distance (FID) score is on the y-axis in the left plot. Negative log-likelihood (NLL) is the y-axis in the right plot. For both plots, compute used for training is on the x-axis and Imagenet 64x64 is the evaluation dataset. Experimental data of scaling behavior obtained from Figure 10 of Nichol & Dhariwal (2021). See Section A.9 for more details.

In Figure 7, we show that BNSL accurately extrapolates the scaling behavior of Diffusion Generative Models of Images from Figure 10 of Nichol & Dhariwal (2021) when Negative Log-likelihood (NLL) or Frechet Inception Distance (FID) score is on the y-axis and compute used for training is on the x-axis; compute is scaled in the manner that is Pareto optimal with respect to the performance evaluation metric on the y-axis.

## A.10 Extrapolation Results for Generative Models of Video

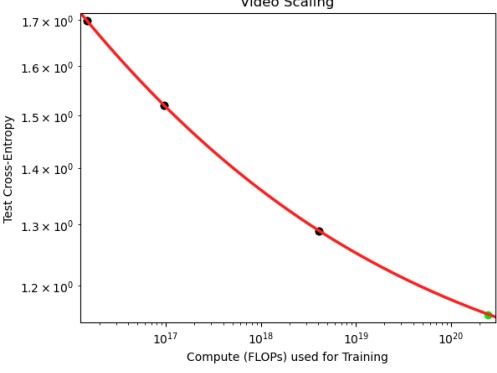

Figure 8: Extrapolation Results of BNSL for scaling behavior of Generative Models of Video. Upstream Test Cross-Entropy is on the y-axis. Videos scraped from the web are the evaluation dataset. During training, compute (used for training autoregressive transformer) on the x-axis is scaled in the manner that is Pareto optimal with respect to the performance evaluation metric on the y-axis. Experimental data of scaling behavior obtained from top right plot of Figure 5 of Henighan et al. (2020). See Section A.10 for more details.

In Figure 8, we show that BNSL accurately extrapolates the scaling behavior of generative models of video. Each frame is downsampled to a pixel resolution of 64x64; each frame is then tokenized via a pretrained 16x16 VQVAE (Van Den Oord et al., 2017) to obtain 256 tokens per frame. 16 consecutive frames are then input to an autoregressive transformer as a length 4096 (16x16x16) sequence. The dataset is 100 hours of videos scraped from the web. See section 2 of Henighan et al. (2020) for more details.

### A.11    EXTRAPOLATION RESULTS WHEN DATA IS PRUNED PARETO OPTIMALLY

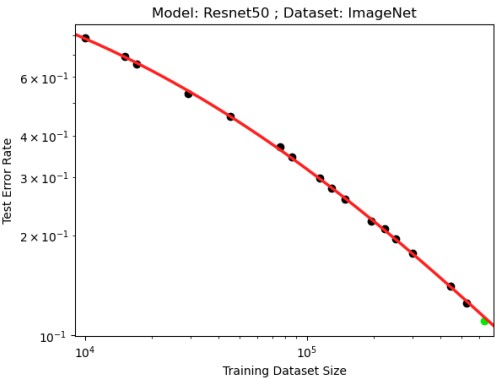

Figure 9: Extrapolation Results of BNSL for scaling behavior when data is pruned Pareto optimally (such that each point along the x-axis uses the subset of the dataset that yields the best performance (y-axis value) for that dataset size (x-axis value)). Experimental data of scaling behavior obtained from Figure 3D of Sorscher et al. (2022). See Section A.11 for more details.

In Figure 9, we show that BNSL accurately extrapolates the scaling behavior when data is pruned Pareto optimally (such that each point along the x-axis uses the subset of the dataset that yields the best performance (y-axis value) for that dataset size (x-axis value)) from Figure 3D of Sorscher et al. (2022).

### A.12    EXTRAPOLATION RESULTS WHEN UPSTREAM PERFORMANCE IS ON THE X-AXIS

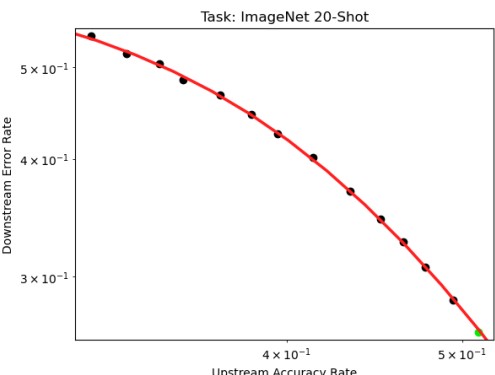

Figure 10: Extrapolation Results of BNSL for scaling behavior when Upstream Performance is on the x-axis and Downstream Performance is on the y-axis. Experimental data of scaling behavior obtained from Figure 5 of Abnar et al. (2021). The upstream task is supervised pretraining of ViT (Dosovitskiy et al., 2020) on subsets of JFT-300M (Sun et al., 2017). The Downstream Task is 20-shot ImageNet classification. See Section A.12 for more details.

In Figure 10, we show that BNSL accurately extrapolates the scaling behavior when upstream performance is on the x-axis and downstream performance is on the y-axis. The upstream task is supervised pretraining of ViT (Dosovitskiy et al., 2020) on subsets of JFT-300M (Sun et al., 2017). The downstream task is 20-shot ImageNet classification. The experimental data of this scaling behavior is obtained from Figure 5 of Abnar et al. (2021).

## A.13 EXTRAPOLATION RESULTS FOR DOWNSTREAM LANGUAGE TASKS WHEN NUMBER OF MODEL PARAMETERS IS ON THE X-AXIS.

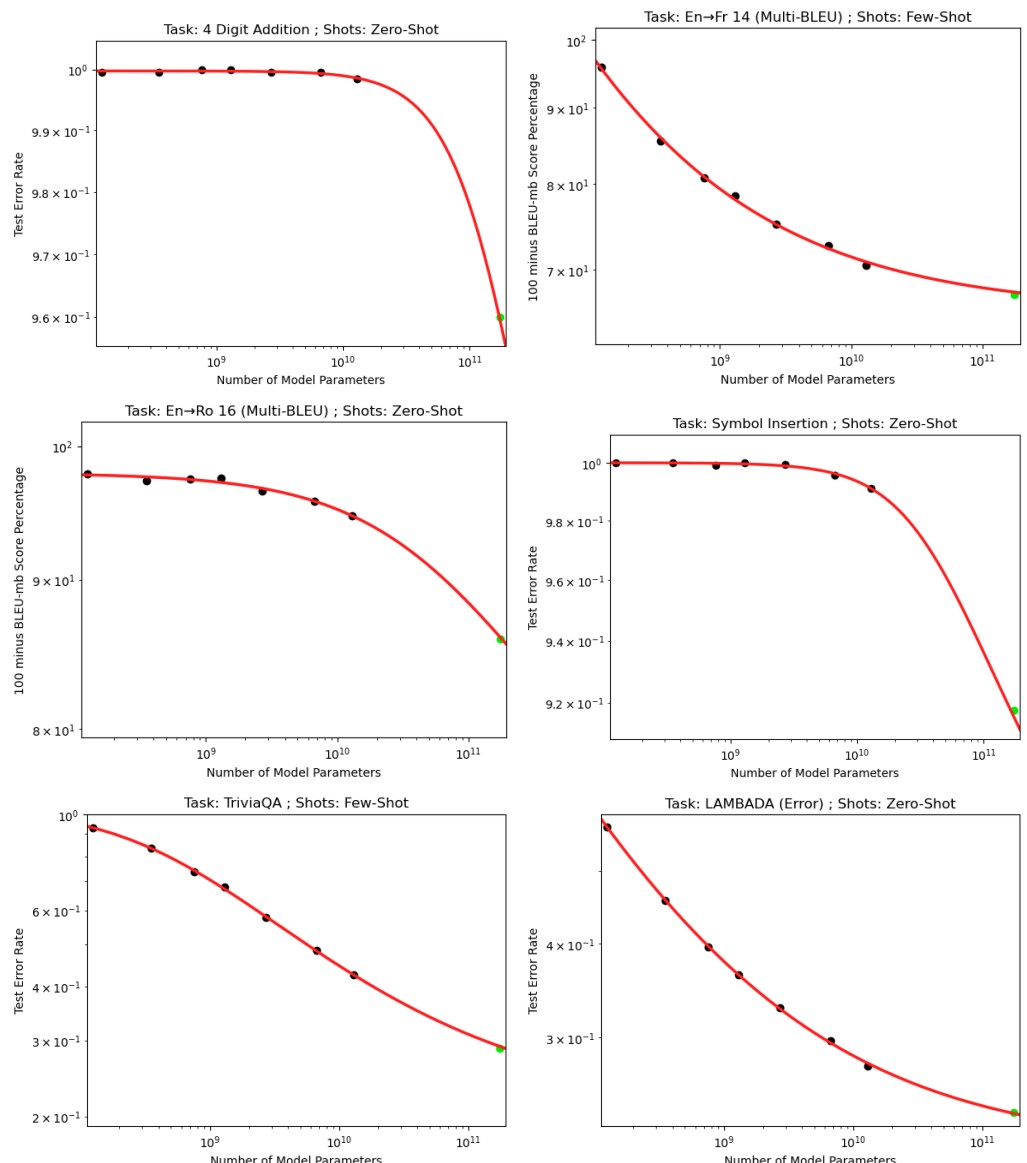

Figure 11: Extrapolation Results of BNSL for Downstream Language Tasks when Number of Model Parameters is on the x-axis. Experimental data obtained from Table H.1 of the GPT-3 arXiv paper (Brown et al., 2020). See Section A.13 for more details.

We find in general for each of every modality that the variance between seeds is higher when number of model parameters is on x-axis (as opposed to e.g. training dataset size on the x-axis). Table H.1 of the GPT-3 arXiv paper (Brown et al., 2020) release includes results for 8 numbers of model parameters. In Figure 11, we include examples of when 8 numbers of model parameters (7 for fitting, and largest held-out to evaluate extrapolation) are sufficient for obtaining accurate downstream extrapolation from BNSL due to variance between seeds being low enough. For many other downstream tasks with number of model parameters on the x-axis, the variance between seeds is much higher such that a number considerably larger than 7 points along the curve is needed to obtain an accurate extrapolation.

## A.14 EXTRAPOLATION RESULTS FOR SPARSE MODELS

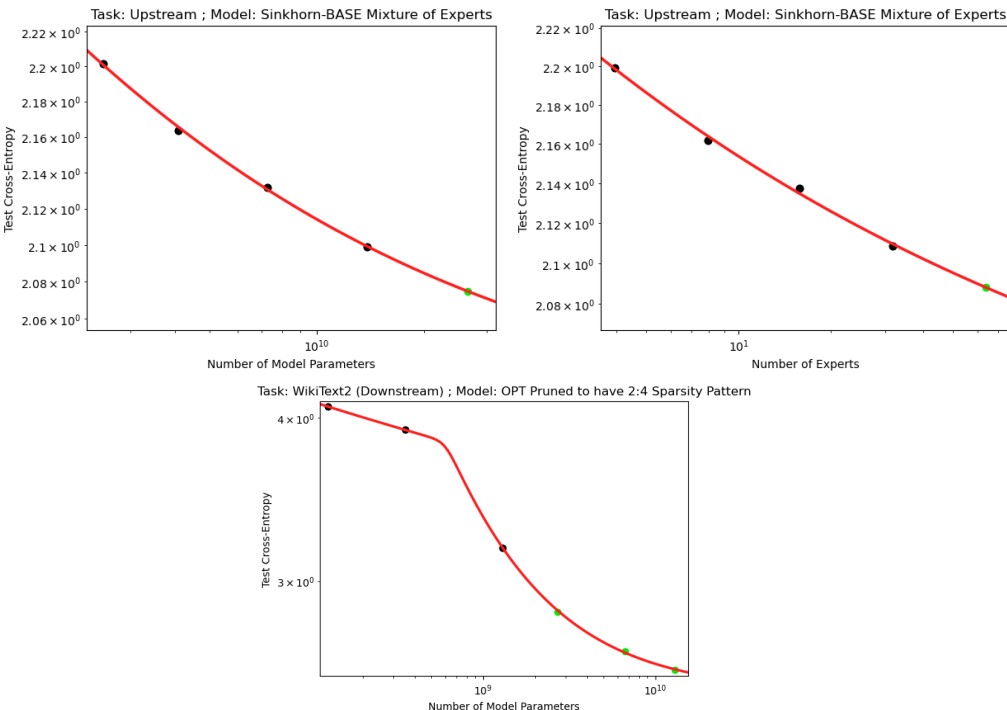

Figure 12: Extrapolation Results of BNSL for Sparse Models. Experimental data of top 2 figures are obtained from Figure 22 of Clark et al. (2022). Experimental data of bottom figure obtained from Figure 1 right of Frantar & Alistarh (2023). The y-axis is Test Cross-Entropy. The x-axis is the number of model parameters that the model contains. See Section A.14 for more details.

In Figure 12, we find BNSL accurately extrapolates the scaling behavior of various sparse models (i.e. sparse, pruned models and sparsely gated mixture-of-expert models).

## A.15 EXTRAPOLATION RESULTS FOR ADVERSARIAL ROBUSTNESS

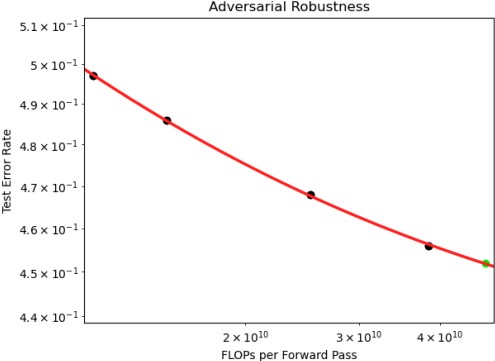

Figure 13: Extrapolation Results of BNSL for Adversarial Robustness. Test Error Rate is on the y-axis. FLOPs of the forward pass of a model of that size is on the x-axis. Experimental data of y-axis is obtained from Table 7 of Xie & Yuille (2020); experimental data of x-axis is obtained from Figure 7 of Xie & Yuille (2020). See Section A.15 for more details.

In Figure 13, we find BNSL accurately extrapolates the scaling behavior of adversarial robustness. The adversarial test set is constructed via a projected gradient descent (PGD) attacker (Madry et al., 2018) of 20 iterations. During training, adversarial examples for training are constructed by PGD attacker of 30 iterations.

### A.16 Extrapolation Results with Finetuning Dataset Size on the X-axis (and also for Computer Programming / Coding)

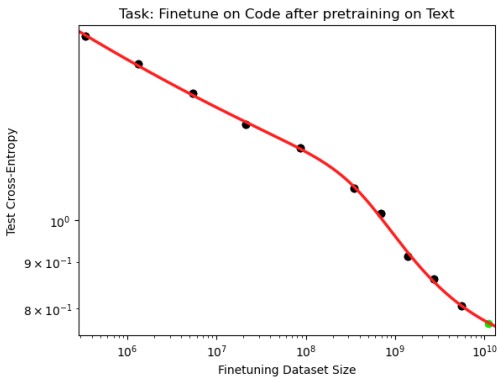

Figure 14: Extrapolation Results of BNSL with Finetuning Dataset Size on the X-axis. Experimental data is obtained from Figure 1 of Hernandez et al. (2021). The figure is of transformer a model that is pretrained on a large amount of mostly English text from the internet and then finetuned on a large amount of python code. The y-axis is Test Cross-Entropy on the distribution of python code. The x-axis is the size (measured in number of characters) of the Finetuning (not pretraining) Dataset. See Section A.14 for more details.

In Figure 14, we find BNSL accurately models and extrapolates the scaling behavior with finetuning dataset size on the x-axis (i.e. model that is pretrained on a large amount of mostly english text from the internet and then finetuned on a large amount of python code).

### A.17 Extrapolation Results for Uncertainty Estimation / Calibration

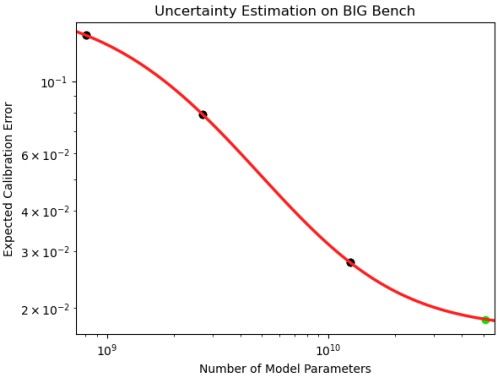

Figure 15: Extrapolation Results of BNSL for Uncertainty Estimation / Calibration. Expected Calibration Error is on the y-axis. Number of Model Parameters is on the x-axis. Experimental data obtained from "Lettered Choices (5-shot)" evaluation protocol plot from Figure 4 right of Kadavath et al. (2022). See Section A.17 for more details.

In Figure 15, we find BNSL accurately extrapolates the scaling behavior of downstream uncertainty estimation / calibration on BIG-Bench (Srivastava et al., 2022).

## A.18 Extrapolation Results for AI Alignment via RLHF

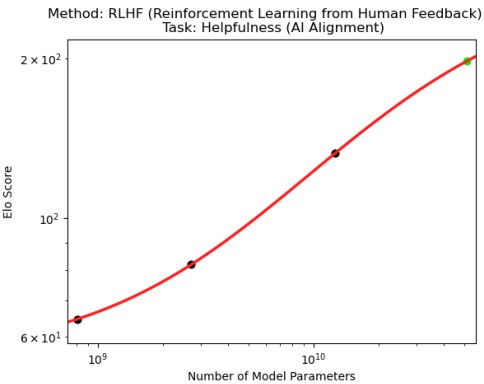

Figure 16: Extrapolation Results of BNSL for Downstream AI Alignment when Number of Model Parameters is on the x-axis. Experimental data obtained from the Static HH RLHF results from Figure 1 of Bai et al. (2022). See Section A.18 for more details.

In Figure 16, we find BNSL accurately extrapolates the scaling behavior of a pretrained language model finetuned (i.e. aligned) via Reinforcement Learning from Human Feedback (RLHF) to be helpful from Figure 1 of Bai et al. (2022). The y-axis is Elo score based on crowdworker preferences. The x-axis is the number of model parameters that the language model contains.

## A.19 Extrapolation Results for Continual Learning (i.e. Catastrophic Forgetting)

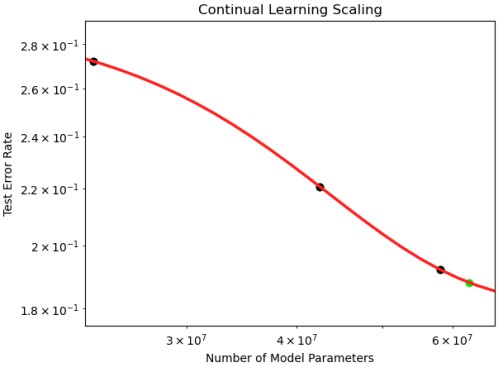

Figure 17: Extrapolation Results of BNSL for Continual Learning (i.e. Catastrophic Forgetting). Experimental data obtained from the Domainnet/Clipart section of the bottom right of Figure 2 of (Ramasesh et al., 2022). X-axis is number of model parameters in the ResNet model. In this setup, model is trained (in sequence, not simultaneously) on task A and then task B. Y-axis is mean of the test error rate on task A and task B. See Section A.19 for more details.

In Figure 17, we find that BNSL accurately extrapolates the scaling behavior of continual learning (i.e. catastrophic forgetting).

## A.20 EXTRAPOLATION RESULTS FOR ROBOTICS (OUT-OF-DISTRIBUTION GENERALIZATION AND IN-DISTRIBUTION GENERALIZATION)

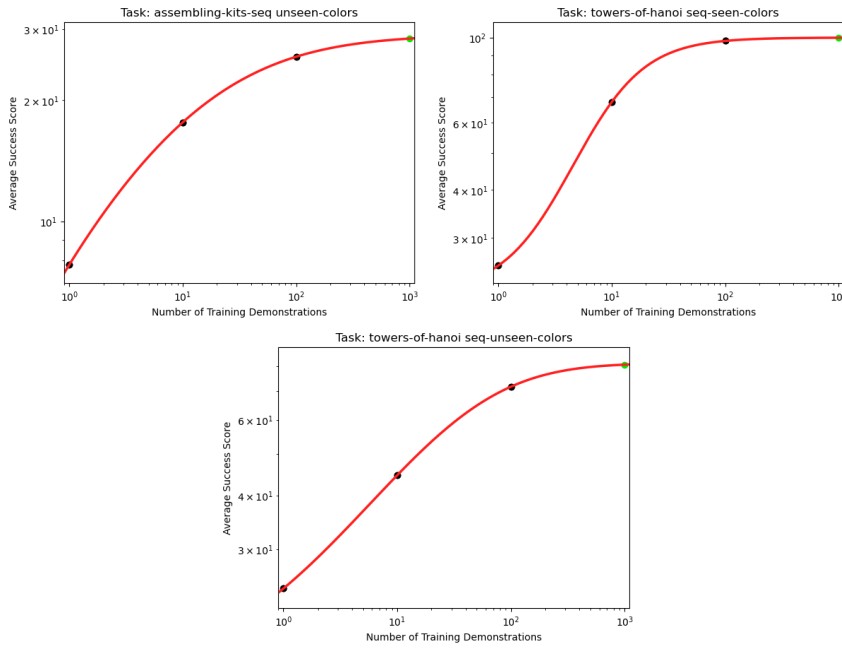

Figure 18: Extrapolation Results of BNSL for Robotic control (and Out-of-Distribution Generalization). Experimental data obtained from the transporter (Zeng et al., 2021) model results from Table 1 of Shridhar et al. (2021). X-axis is number of training demonstrations. Y-axis is task success score (mean percentage) obtained via 100 evaluations. See Section A.20 for more details.

In Figure 18, we find BNSL accurately extrapolates the scaling behavior of a transporter (Zeng et al., 2021) model trained via imitation learning to do robotic control tasks. Plots with "unseen-colors" in the plot title evaluate on a test set that contains colors that are unseen (i.e. out-of-distribution) with respect to the training set. Plots with "seen-colors" in the plot title evaluate on a test set that contains colors that are seen (i.e. in-distribution) with respect to the training set.

## A.21 EXTRAPOLATION RESULTS FOR OUT-OF-DISTRIBUTION DETECTION

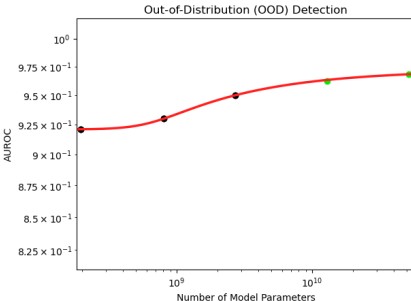

Figure 19: Extrapolation Results of BNSL for Out-of-Distribution Detection. Number of model parameters is on the x-axis. Y-axis is AUROC. Experimental data obtained from the Outlier Exposure results from Figure 23 of Bai et al. (2022) when exposed to 30 outlier examples. See Section A.21 for more details.

In Figure 19, we find BNSL accurately extrapolates the scaling behavior of Out-of-Distribution Detection performance.

## A.22 EXTRAPOLATION RESULTS FOR MOLECULES

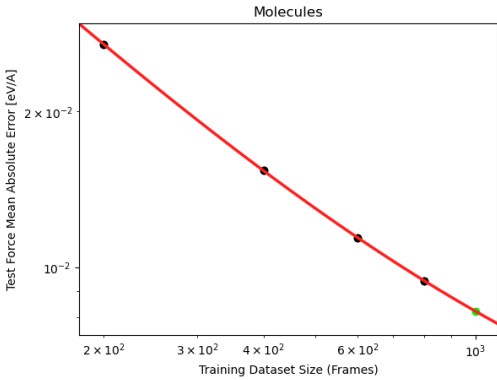

Figure 20: Extrapolation Results of BNSL for Molecules. Experimental data obtained from the "NequIP L=3" results for the aspirin molecule in MD-17 of Figure 8 of the arXiv version of Batzner et al. (2022). Y-axis is the test force mean absolute error [eV/A]. X-axis is the training dataset size (frames). See Section A.22 for more details.

In Figure 20, we find BNSL accurately extrapolates the scaling behavior of Neural Equivariant Interatomic Potentials (NequIP) graph neural networks (Batzner et al., 2022) trained via minimizing the weighted sum of energy and a force loss terms in order to predict the forces of molecules.

## A.23 EXTRAPOLATION RESULTS FOR MATH WORD PROBLEMS

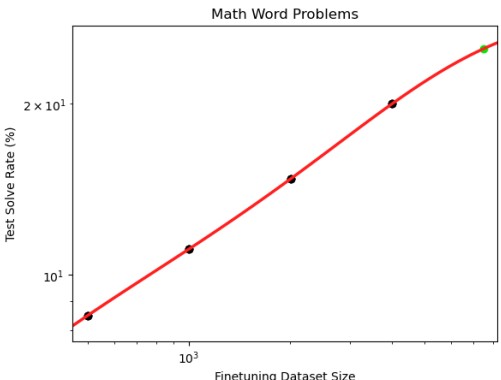

Figure 21: Extrapolation Results of BNSL for Math Word Problems. Experimental data obtained from the 12 billion parameter model results in Figure 2 left of Cobbe et al. (2021a). Y-axis is the test solve rate. X-axis is the finetuning dataset size. See Section A.23 for more details.

In Figure 21, we find BNSL accurately extrapolates the scaling behavior of large language models finetuned to solve math word problems.

### A.24 EXTRAPOLATION RESULTS FOR DOWNSTREAM PERFORMANCE OF MULTIMODAL CONTRASTIVE LEARNING (I.E. NON-GENERATIVE UNSUPERVISED LEARNING)

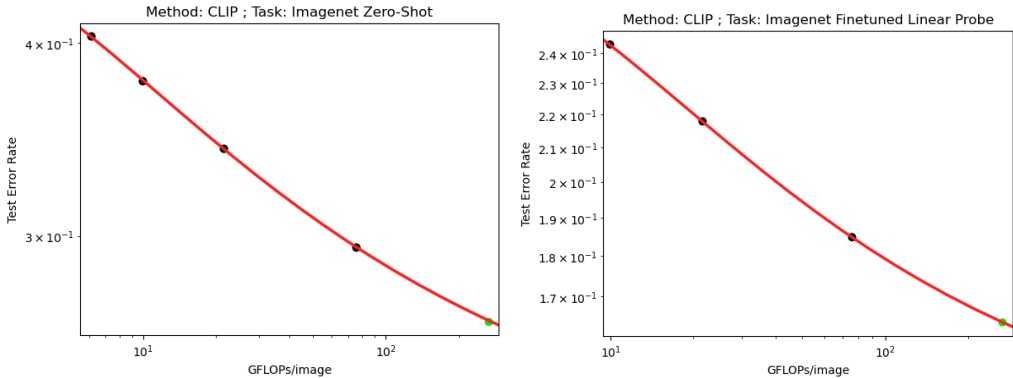

Figure 22: Extrapolation Results of BNSL for Downstream Performance of Multimodal Contrastive Learning (i.e. Non-Generative Unsupervised Learning). Experimental data of scaling behavior obtained from Table 10 and Table 11 in arXiv version of Radford et al. (2021). The upstream task is "Contrastive Image Language Pretraining" (a.k.a. CLIP) of ResNets on a training dataset consisting of hundreds of millions of image-text pairs. The x-axis is GFLOPs/image (GigaFLOPs/image) of the forward-pass of model. The Downstream Task is ImageNet classification (i.e. the y-axis of plot). The y-axis of left plot is Zero-Shot Downstream. The y-axis of right plot is performance of model with finetuned linear probe on it. See Section A.24 for more details.

In Figure 23, we show that BNSL accurately extrapolates the scaling behavior of the Downstream Performance of Multimodal Contrastive Learning (i.e. Non-Generative Unsupervised Learning).

### A.25 EXTRAPOLATION RESULTS FOR DOWNSTREAM PERFORMANCE ON AUDIO TASKS

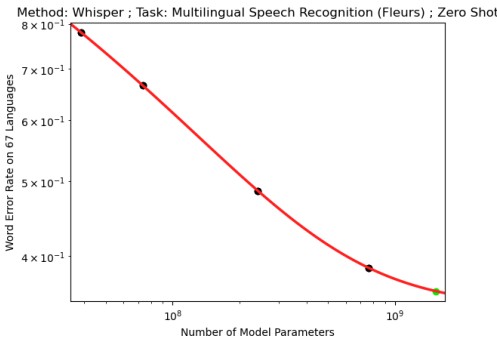

Figure 23: Extrapolation Results of BNSL for Downstream Audio Tasks when Number of Model Parameters is on the x-axis. Experimental data obtained from the second plot of Figure 6 of Whisper paper (Radford et al., 2022). The downstream task in the plot is downstream zero shot multilingual speech recognition performance on the FLEURS dataset of "Whisper" speech recognition model pretrained on a dataset of 681,070 hours of audio. See Section A.25 for more details.

In Figure 23, we show that BNSL accurately extrapolates the scaling behavior of the Downstream Performance on Audio Tasks.

## A.26  Plots of BNSL Extrapolations on Scaling Laws Benchmark of Alabdulmohsin et al. (2022)

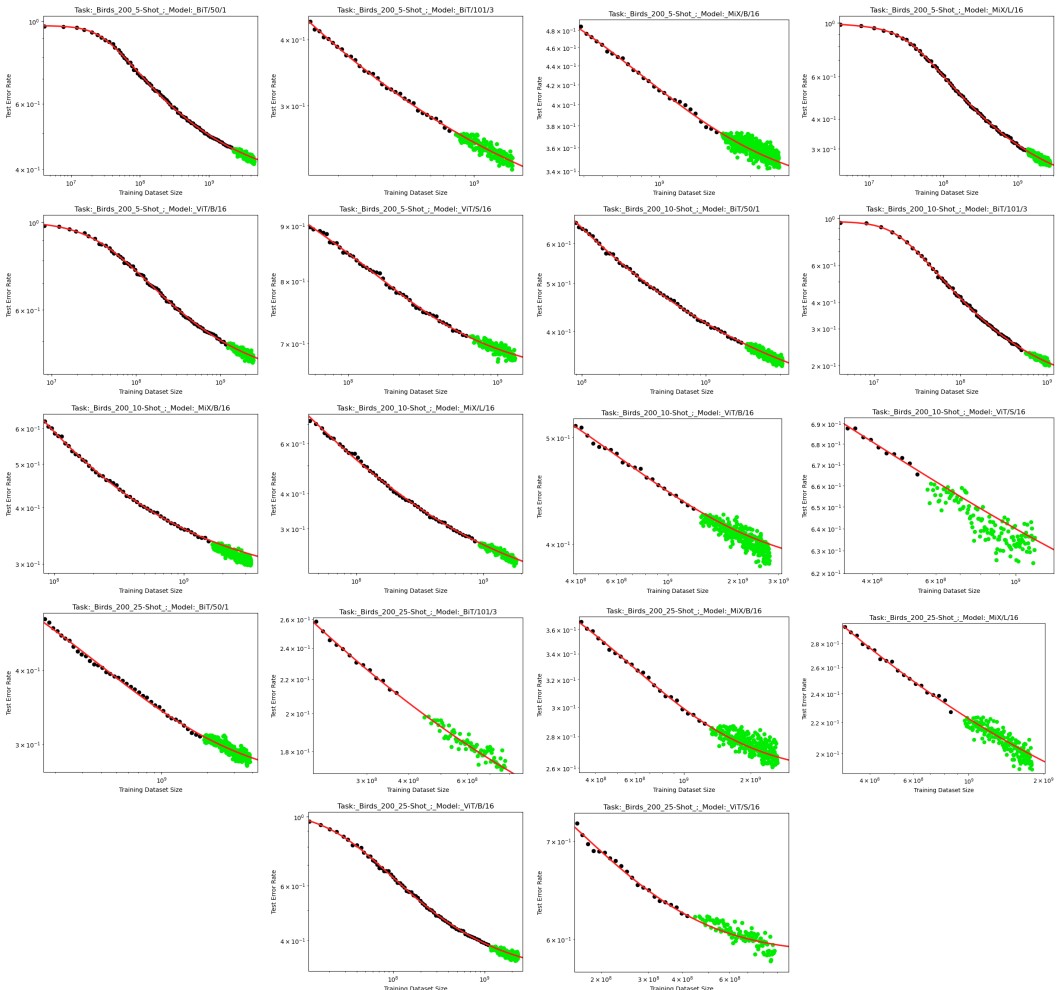

Figure 24:  Extrapolation Results of BNSL on Downstream Birds 200. X-axis is pretraining dataset size. See Section 5.1 for more details.

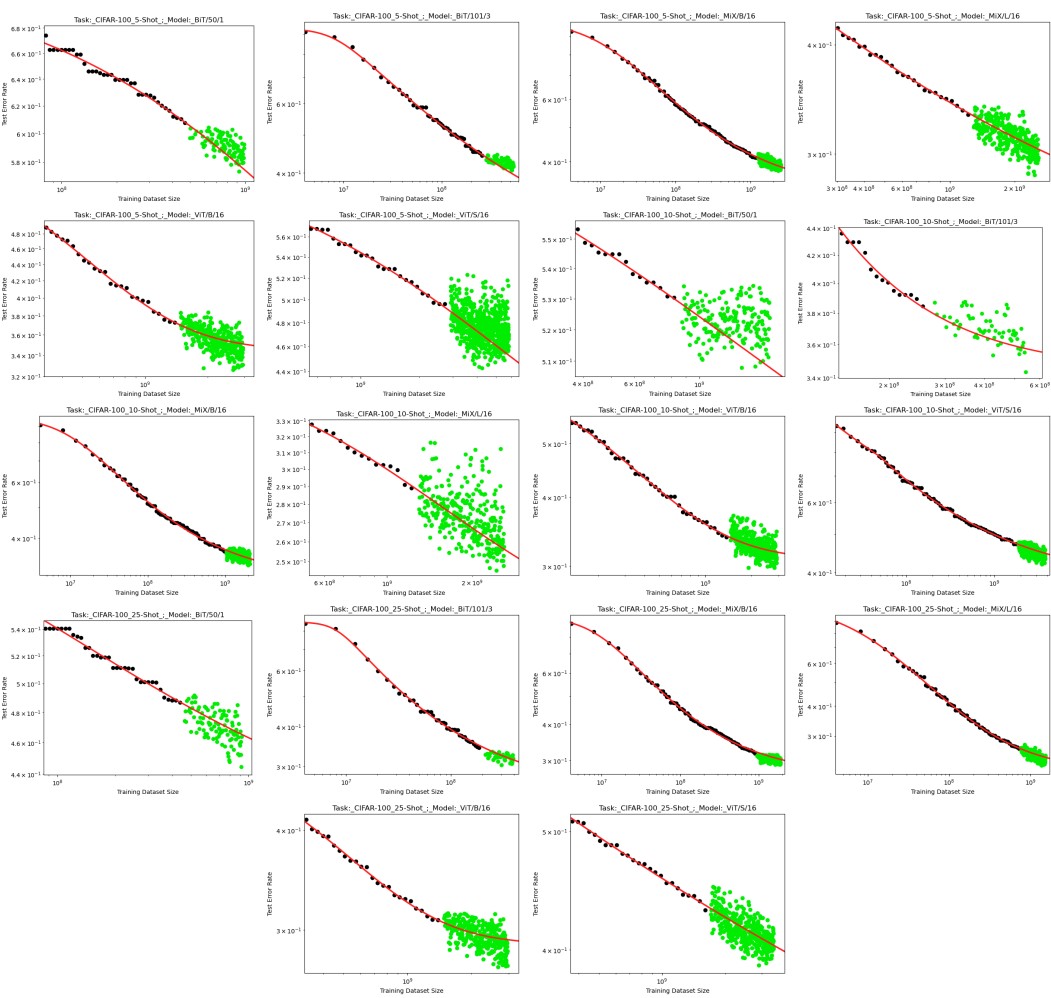

Figure 25: Extrapolation Results of BNSL on Downstream CIFAR-100. X-axis is pretraining dataset size. See Section 5.1 for more details.

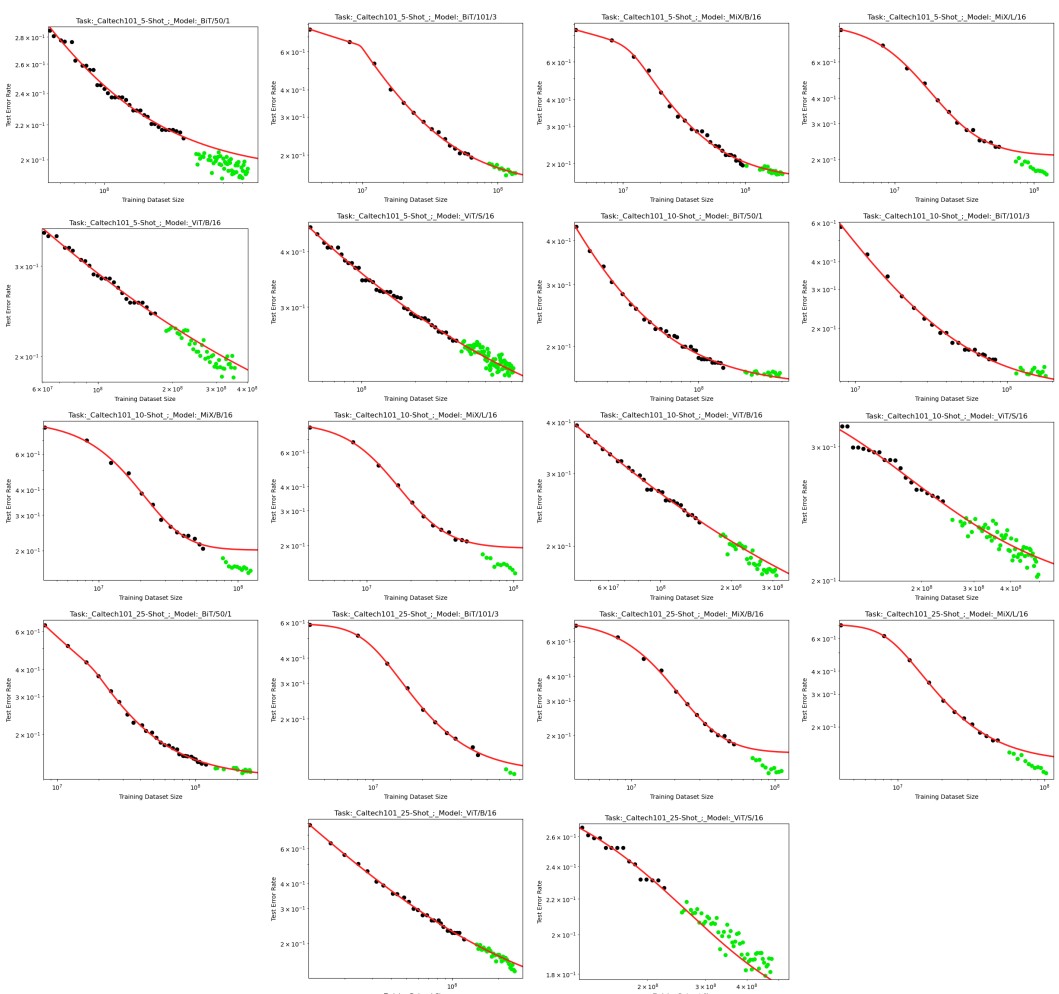

Figure 26: Extrapolation Results of BNSL on Downstream Caltech101. X-axis is pretraining dataset size. See Section 5.1 for more details. From eyeballing, we think the subset of Caltech101 with unsatisfactory extrapolations has unsatisfactory extrapolations due to the maximum (along the x-axis) of the black point used for fitting being near or before a break; this is accentuated by not having enough points for fitting for the SciPy fitter to be able to determine whether the break is an actual break or just noisy deviation. **See Section 6** for more details on this explanation.

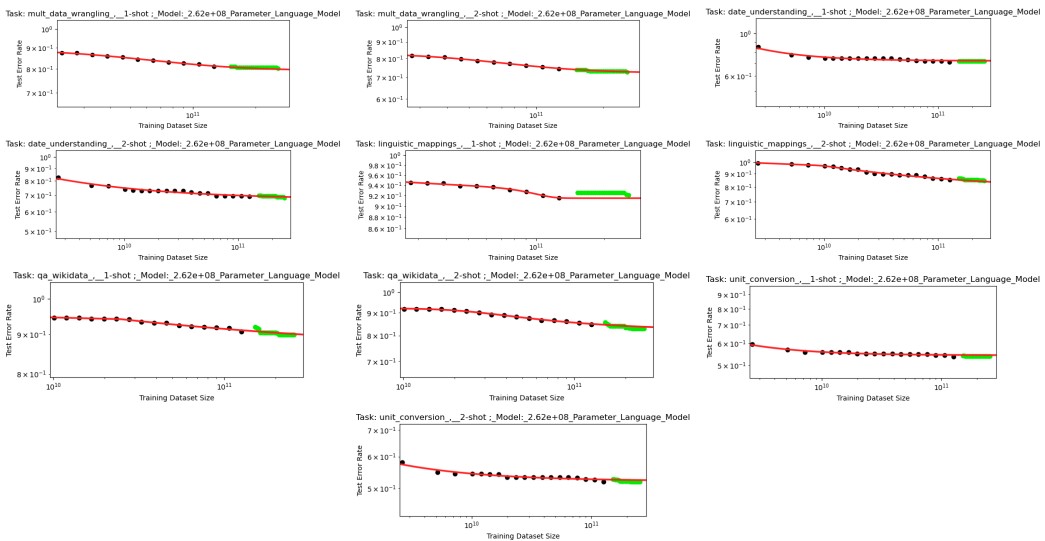

Figure 27: Extrapolation Results of BNSL on Downstream BIG-Bench (BB). X-axis is pretraining dataset size. See Section 5.2 for more details. From eyeballing, we think the subset of BIG-Bench with unsatisfactory extrapolations has unsatisfactory extrapolations due to the maximum (along the x-axis) of the black point used for fitting being near or before a break; this is accentuated by not having enough points for fitting for the SciPy fitter to be able to determine whether the break is an actual break or just noisy deviation. **See Section 6** for more details on this explanation.

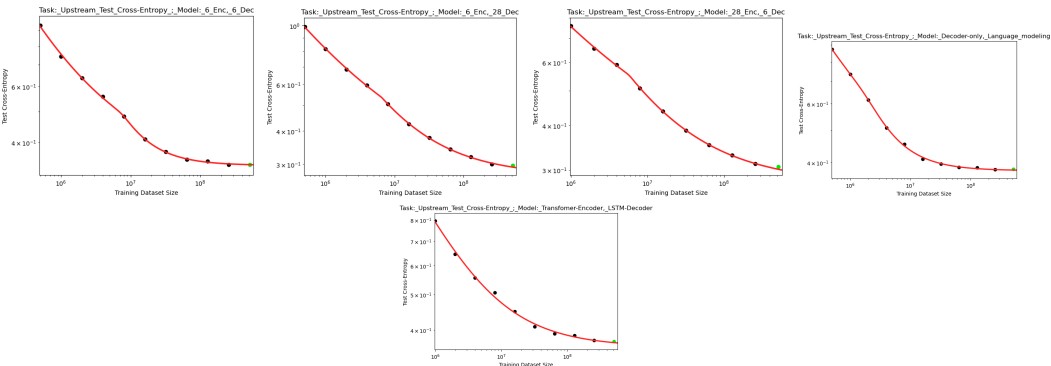

Figure 28: Extrapolation Results of BNSL on Neural Machine Translation (NMT). See Section 5.2 for more details.

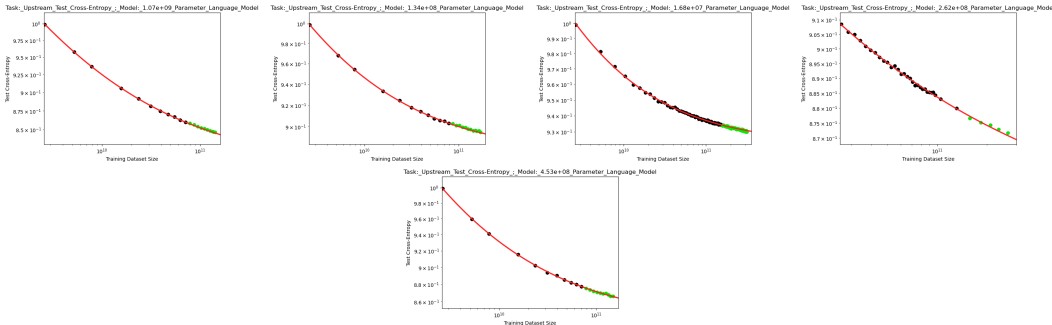

Figure 29: Extrapolation Results of BNSL on Language Modeling (LM). See Section 5.2 for more details.

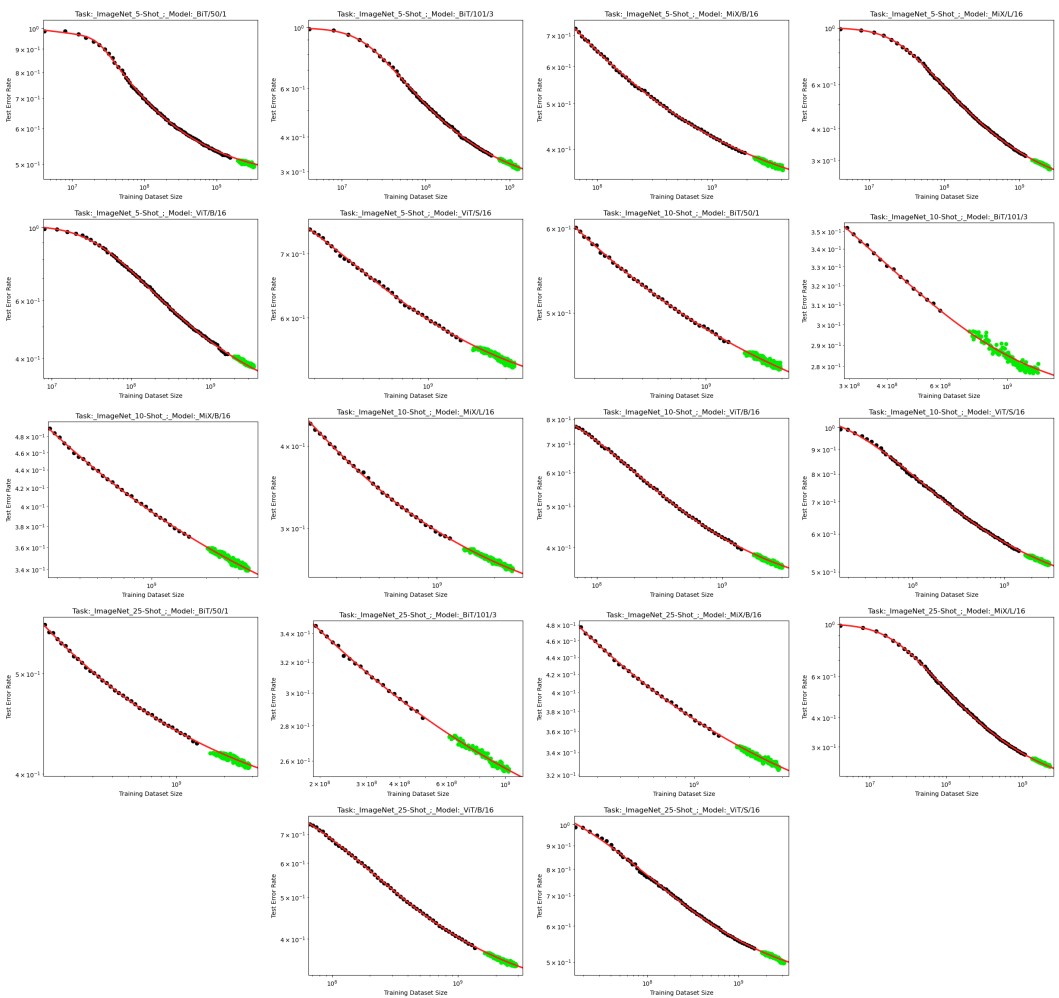

Figure 30: Extrapolation Results of BNSL on Downstream ImageNet. X-axis is pretraining dataset size. See Section 5.1 for more details.

