# OpenReview forum: "Broken Neural Scaling Laws"
_ICLR.cc/2023/Conference — ICLR 2023 poster_

### Official Review · Reviewer_hC3X · 2022-10-18

**Confidence:** 3
**Correctness:** 4
**Technical Novelty And Significance:** 3
**Empirical Novelty And Significance:** 3
**Recommendation:** 8

**Clarity, Quality, Novelty And Reproducibility:**

The paper is very clear and the language is standard language in academia.

The mathematical language of the paper could have been more rigorous. There is not many propositions and theorems that define the theoretical analysis the scalings, the formulation limitations, or boundaries of the problem. The quality of the appendices are low because it lacks discussion and is mainly focused on experimental plots.

The experimental aspect of the paper is high quality as it is experimented on multiple task settings and famous datasets. In comparison to Alabdulmohsin et al (2022) the novelty is less than average as proposing a new functional form for scaling neural networks has been around in multiple previous works. The mentioned paper Also in comparison, the suggested formulation by Hernandez et al. (2021) proposes more specific case of fine-tuning and limited to dataset size.





**Strength And Weaknesses:**

The strengths of this paper include but not limited to clarity of the idea presented in presenting a formula to model scaling behavior neural networks in learning tasks is very clear and somehow intuitive idea. Second, this paper is extremely concise and the basic mathematical idea behind the paper sounds solid and intuitive. Another strength is significant enhancement in the performance of the scaling law in neural networks, at least compared to other M1 to M4 methods.

Some of the weaknesses are presented in the A.3 section. Although the results show stronger performance compared to other proposed formulations, the figures show significant deviations in performance. This is not necessarily a weakness however it is worthy of addressing in the paper. In addition, the performance for cases with more monotonic extrapolation sounds reasonable in some simpler cases however in more complex nonlinear cases, the duration of predictions sound too short to give any notion of the performance. My concern is the number of test points has been cherry picked based on the strength of this approach. In addition, the practicality of modeling scaling behavior is way more valuable when we can predict a certain behavioral patterns sooner than later. In other words, modeling with fewer datapoints and extrapolating on more points than is presented. Finally, I am curious how does the scaling formula is compared to other methods aside from direct formulations such as predictive models and machine learning extrapolations. The reason for my notion is my observation in the A.3 that this formulation misses some basic locally linear extrapolations in some cases. For example, in Fig.6 the results sounds like a simple linear regression in log-log plot models the scaling behavior better and the BNSL actually is underestimating the scaling behavior. This also intensifies my previous concern that longer extrapolation horizon might show even higher and more staggering deviations between results and the BNSL prediction.

**Summary Of The Paper:**

The authors propose using an functional form of smoothly broken power law namely broken neural scaling laws (BNSL) for extrapolating and generalizing to model scaling behaviors across diverse set of upstream and downstream tasks such zero-shot, prompted, and fine-tuned settings of unsupervised language, vision, reinforcement learning, and arithmetic. The results shows that using BNSL is significantly more accurate in extrapolating the scaling laws based on dataset size and model complexity. These benefits are driven from two important characteristics of BNSL to model non-monotonic transitions in scaling behavior (e.g in double descent) or and inflection points (e.g. in delayed and sharp transitions in arithmetic).

**Summary Of The Review:**

The paper has clear and solid and intuitive idea and observations, however, limited presentation to support the theory behind the scaling law. The presentation is not rigorous in terms of theoretical foundation and too much dependent on the intuitive explanation of the formula's property. The experimental results can be improved and acceptable by including longer extrapolation horizon to show the true capability of this functional form for more sophisticated extrapolation (not just more sophisticated modeling) and this functional form can be compared to non-functional scaling laws (the reviewer were not able to find precedent to non-functional forms of scaling laws in recent literature). To the reviewer, proposing one functional form for all the task settings and networks is not convincing. There might be the case that different scaling formulas can be applied to different neural networks and deeper analysis of the scaling formula in combination with the network properties gives better view of the scaling behavior.

---

> ### Author Response · Authors · 2022-11-15
> **What are examples of "non-functional" forms of scaling laws?**
>
> Hi Reviewer hC3X,
>
> We are not sure what "non-functional" forms you are referring to when you ask about "non-functional" forms of scaling laws. What are examples of "non-functional" forms of scaling laws that you ask about?

---

> > ### Comment · Reviewer_hC3X · 2022-11-21
> > **Explaining non-functional forms**
> >
> > By non-functional I meant learning equations for extrapolation. Although after reviewing multiple papers in literature (e.g. Learning Equations for Extrapolation) I believe this request is out of scope for this paper. The coverage of functional forms of scaling laws are satisfactory in this paper.

---

> ### Author Response · Authors · 2022-11-19
> **Reply to Reviewer hC3X (1/2)**
>
> We thank the reviewer for useful feedback, and attempt to clarify below all the concerns raised.
>
> > “Although the results show stronger performance compared to other proposed formulations, the figures show significant deviations in performance. This is not necessarily a weakness however it is worthy of addressing in the paper.
> “... this formulation misses some basic locally linear extrapolations in some cases. For example, in Fig.6 the results sounds like a simple linear regression in log-log plot models the scaling behavior better and the BNSL actually is underestimating the scaling behavior.
>
> Regarding the  concern that our functional form “misses some basic locally linear extrapolations” “for example, in Fig.6”, please note that  such unsatisfactory extrapolations seen in  several  plots in the appendix (e.g. on a subset of Caltech101 in Fig.6) occurs likely due to the fact that  the majority (along the x-axis) of the black points used for fitting is located near or just before the break; this is accentuated by not having enough data points for the SciPy fitter to be able to determine whether this is truly a break in the underlying performance curve or just a noisy deviation. We discuss this limitation in Section 5.6 titled “The Limit of the Predictability of Scaling Behavior”. Also, please note that  this limitation is not specific to our functional form. As we can see  in Fig. 21, 22, and 23 of https://arxiv.org/pdf/2209.06640v2.pdf#page=25 (Alabdulmohsin et al (2022)), other  functional forms proposed as models of  neural scaling behavior are also unable to yield satisfactory extrapolations in such scenarios,  because the information needed to obtain an accurate extrapolation is just not present in the data in those cases. You suggest that a “simple linear regression in log-log plot models the scaling behavior” in such scenarios; however, functional form M1 (which is  y=bx^c ) is exactly a simple linear regression in log-log plot, and, as can be seen in Fig. 21, 22, and 23 of https://arxiv.org/pdf/2209.06640v2.pdf#page=25 (Alabdulmohsin et al (2022)), M1 neither fits nor extrapolates well such scaling behaviors; it’s also worth noting that a BNSL with the number of breaks n set to zero is M1 plus a constant (y= a+bx^c).
>
> > “ My concern is the number of test points has been cherry picked based on the strength of this approach”
>
> We do not cherrypick the number of test points in any of the experimental data provided by Alabdulmohsin et al (2022); we use all the test points that Alabdulmohsin et al (2022) provided as is for evaluating extrapolation. The experimental data is provided here by Alabdulmohsin et al (2022) if you wish to check:
> https://github.com/google-research/google-research/tree/master/revisiting_neural_scaling_laws/data
>
> > “In addition, the practicality of modeling scaling behavior is way more valuable when we can predict a certain behavioral patterns sooner than later. In other words, modeling with fewer datapoints and extrapolating on more points than is presented.” “This also intensifies my previous concern that longer extrapolation horizon might show even higher and more staggering deviations between results and the BNSL prediction.” “The experimental results can be improved and acceptable by including longer extrapolation horizon to show the true capability of this functional form for more sophisticated extrapolation (not just more sophisticated modeling)”
>
> With regards to longer extrapolation horizon, we now include accurate extrapolation to scales that are order(s) of magnitude larger than the maximum (along the x-axis) of the points used for fitting in Appendix A.10 titled “Extrapolation to Scales that are Order(s) of Magnitude larger than the maximum (along the x-axis) of the points used for fitting”.
> In this section of the Appendix, we find that BNSL yields accurate extrapolations to scales 2 order(s) of magnitude larger than the maximum (along the x-axis) of the points used for fitting and that BNSL’s extrapolations in this (2 order(s) of magnitude) setting are considerably more accurate than all the other functional forms (M1, M2, M3, and M4).
>
> > “Finally, I am curious how does the scaling formula is compared to other methods aside from direct formulations such as predictive models and machine learning extrapolations.” “this functional form can be compared to non-functional scaling laws (the reviewer were not able to find precedent to non-functional forms of scaling laws in recent literature).”
>
> We asked reviewer h3CX for clarification in https://openreview.net/forum?id=sckjveqlCZ&noteId=8DZNKqeDlA and reviewer h3CX has since confirmed that “non-functional” form comparisons are out of scope for this submission and that the coverage provided by the current comparison between the functional forms in our submission is satisfactory.

---

> > ### Author Response · Authors · 2022-11-19
> > **Reply to Reviewer hC3X (2/2)**
> >
> > > “The paper has clear and solid and intuitive idea and observations, however, limited presentation to support the theory behind the scaling law.” “The presentation is not rigorous in terms of theoretical foundation and too much dependent on the intuitive explanation of the formula's property.” “The mathematical language of the paper could have been more rigorous. There is not many propositions and theorems that define the theoretical analysis the scalings, the formulation limitations, or boundaries of the problem.” “The quality of the appendices are low because it lacks discussion and is mainly focused on experimental plots.”
> >
> > Re: mathematical rigor and discussion, we provide a formal discussion of the functional form, e.g. see Appendix A1. Regarding theoretical analysis of behaviors of large deep neural networks, it can be quite nontrivial, and theoretical derivation of the scaling function “from the first principles” is clearly hard (i.e. it is still not known why scaling behaviors of artificial neural networks often follow/involve power laws in such a wide variety of settings despite power law scaling behavior of multilayer neural networks first being observed over 20 years ago in https://papers.nips.cc/paper/1993/file/1aa48fc4880bb0c9b8a3bf979d3b917e-Paper.pdf ). Note that the field of neural scaling laws is inherently empirical by its nature, precisely because performing (useful) theoretical analysis of very large, deep, and complex network’s performance (and other behaviors) becomes exceedingly (some would say prohibitively) complicated. Thus, the empirical science approach to neural performance is necessary, akin to studies of complex network behavior in other fields, e.g. statistical physics ( e.g. see 1:16:39 onward of  https://youtu.be/CR45mBkSH7g?t=4599 ). Of course, a deeper theoretical understanding of the complex network mechanisms and data properties underlying neural net behavior remain an important, but challenging open question.
> >
> > > In comparison to Alabdulmohsin et al (2022) the novelty is less than average as proposing a new functional form for scaling neural networks has been around in multiple previous works. The mentioned paper Also in comparison, the suggested formulation by Hernandez et al. (2021) proposes more specific case of fine-tuning and limited to dataset size.
> >
> > We agree that several functional forms (e.g. M1-M4) were proposed before; however, none of them extrapolate scaling behavior as accurately on such a wide variety of datasets and tasks.

---

### Official Review · Reviewer_72f9 · 2022-10-23

**Confidence:** 3
**Correctness:** 4
**Technical Novelty And Significance:** 3
**Empirical Novelty And Significance:** 3
**Recommendation:** 8

**Clarity, Quality, Novelty And Reproducibility:**

Overall the paper is clearly written and the functional form has intuitive value that addresses weaknesses of previous forms. The paper also includes details on how to train the predictor.

It might be helpful to provide more intuition on equation (1), e.g., with another equation that describes the relationship in a log-log form.

**Strength And Weaknesses:**

Strengths:
- The paper has many extensive experiments and the function is intuitive and can be easily implemented

Weaknesses:
- The authors remark on a limitation which I wish that they numerically investigated. Just how many points do we need to fit this function? Simple neural scaling laws can be fit with even less than 10 points, whereas since this function has lots of parameters, I suspect it can be challenging. It would be nice to run some ablations on this, especially since it is an easy test.

- How do we determine $n$ when forming a BNSL? This seems to be a parameter that must be specified a priori and not learned. Is there intuition in terms of how we can go about selecting it? Is there a consequence for selecting n too large or too small?

**Summary Of The Paper:**

This paper proposes a new functional form for neural scaling laws that corrects for two weaknesses in previous functions: (1) that they could only model strict monotonic behaviour, and (2) they could not express inflection points. The function is intuitive and well-designed and outperforms previous scaling law functions on learning curve fitting, sometimes by orders of magnitude. These are evaluated on many vision and language tasks to demonstrate better accuracy than other methods.


**Summary Of The Review:**

Overall the paper presents a simple, intuitive function to train and demonstrates its value extensively on both vision and language tasks. Although the paper identifies limitations, it would be nice to also evaluate the degree of the limitations, especially since they are not too hard to test.

---

> ### Author Response · Authors · 2022-11-19
> **Reply to Reviewer 72f9**
>
> Thank you for your review and comments, we are glad that you found the contribution of our paper interesting and significant.
>
> > 1) “Just how many points do we need to fit this function?”
>
> We assume you mean “Just how many points do we need to fit this function” well enough to get an accurate extrapolation. When the noisy deviation between seeds is low, we are able to obtain accurate extrapolations with as low as 6 points / seeds in practice. When the noisy deviation between seeds is extremely high (e.g. during the most extreme break of 4 digit addition), we need hundreds of points / seeds to fit this function well enough to get an accurate extrapolation of 4 digit addition when only fitting using points whose maximum (along the x-axis) is near the break; (the variance between seeds is highest near breaks in which the change in slope (on a log-log plot) is large and fast). Most of the time in practice, the number of seeds / points needed “to fit this function” well enough to get an accurate extrapolation is a number between 6 and 25.
>
> > 2) “Although the paper identifies limitations, it would be nice to also evaluate the degree of the limitations, especially since they are not too hard to test.” “Simple neural scaling laws can be fit with even less than 10 points, whereas since this function has lots of parameters, I suspect it can be challenging. It would be nice to run some ablations on this, especially since it is an easy test.”
>
> We had evaluated limitations in Section 5.6. Now, we include additional ablations (to probe any limitations) in Appendix A.10 titled “Extrapolation to Scales that are Order(s) of Magnitude larger than the maximum (along the x-axis) of the points used for fitting”.
>
> > 3) How do we determine n when forming a BNSL? This seems to be a parameter that must be specified a priori and not learned. Is there intuition in terms of how we can go about selecting it? Is there a consequence for selecting n too large or too small?
>
> If n is too small, one underfits. If n is too large (and the number of points for fitting is too small), one overfits. The simplest way to go about determining n when forming a BNSL is to first try fitting a 0 break BNSL. If that 0 break BNSL is underfitting, then try fitting a 1 break BNSL. If that 1 break BNSL is underfitting, then try fitting a 2 break BNSL. And so on until one is no longer underfitting by a significant amount. Most of the time for most of the scaling behaviors that people care about in practice, the number of breaks needed to accurately model and extrapolate a scaling behavior is 1 break. An example of a scaling behavior in which 2 breaks is required is double descent in figure 3 left of the rebuttal version of paper: The first break models the first non-monotonic transition, and the second break models the second non-monotonic transition.
>
> > 4) It might be helpful to provide more intuition on equation (1), e.g., with another equation that describes the relationship in a log-log form.
>
> We now provide such intuition in Appendix A.1 via a mathematical analysis (i.e. equation (3) which describes the relationship in a log-log form) and explanation of why BNSL is a smoothly connected piecewise (approximately) linear function on a log-log plot.

---

### Official Review · Reviewer_TK8J · 2022-10-24

**Confidence:** 3
**Correctness:** 3
**Technical Novelty And Significance:** 3
**Empirical Novelty And Significance:** 3
**Recommendation:** 6

**Clarity, Quality, Novelty And Reproducibility:**

Clarity: The writing is clear and easy to follow.

Quality: The proposed functional is carefully tested on many different settings. The quality of the numerical study is high.

Novelty: The broken power law functional is novel in the studying of neural scaling law. It has been used in other fields like astrophysics.

Reproducibility: I did not check the reproducibility of the experiments.

**Strength And Weaknesses:**

Strength:

The broken power law functional can fit the performance of neural networks over the change of its size more accurately. It can work on more networks, problems, and metric of performances.

Weaknesses:

Compared with existing power laws, the proposed functional has more free parameters. Hence, it is not surprising that it can fit the scaling behavior more accurately. To show the power of this proposed functional, the authors need to show that it can give more accurate predictions to the performance of neural networks than the power laws, or the prediction works in a larger range of network size (or other factors). As shown in the numerical results in the appendix, it seems in many cases the functional does not provide good prediction to the green points even though those points only span less than one order of magnitude in the horizontal axis.

**Summary Of The Paper:**

This paper proposes a piecewise defined "broken power law functional" that can accurately fit the performance of neural networks when quantities like the amount of compute used for training, number of model parameters, or training dataset size varies across several orders of magnitudes. The accuracy of the functional is verified on many different tasks. Compared with existing power laws, the proposed functional can fit non-monotonic behaviors like the double-descent phenomenon

**Summary Of The Review:**

The paper proposes a new functional that can more accurately fit the scaling behavior of neural networks. The concern is that the accuracy of the functional comes from the flexibility provided by undetermined parameters, rather than characterizing the true pattern behind the scaling behaviors.

---

> ### Author Response · Authors · 2022-11-19
> **Reply to reviewer TK8J**
>
> We thank the reviewer for the comments, and provide a detailed response that addresses reviewer’s concerns.
>
> > 1) “The concern is that the accuracy of the functional comes from the flexibility provided by undetermined parameters, rather than characterizing the true pattern behind the scaling behaviors.” “Compared with existing power laws, the proposed functional has more free parameters. Hence, it is not surprising that it can fit the scaling behavior more accurately. To show the power of this proposed functional, the authors need to show that it can give more accurate predictions to the performance of neural networks than the power laws, or the prediction works in a larger range of network size (or other factors)"
>
> We agree that BNSL should “give more accurate predictions to the performance of neural networks than the power laws”, which is why 100% of the results presented in the rebuttal version of the paper are extrapolations from small scales to larger scales.  Note that it was already the case in the original version of our paper that the most thorough quantitative results, summarized in Table 2, show that our model typically extrapolates better than any of the four previous functional forms considered by Alabdulmohsin et al, 2022 [1] (note that the basic power laws are included in that comparison).
>
> While the extrapolations using our functional form is not immaculate in some cases, the results in Table 2 clearly demonstrate that the other functional forms tend to be significantly worse. To further clarify this point, we now added a visualization of Table 2 results as plots in the Appendix, in section titled “Plots of All Extrapolations (M1, M2, M3, M4, and BNSL) on Scaling Laws Benchmark of Alabdulmohsin et al, 2022” [1].
>
> Hope this  clarifies any possible confusion and addresses  the concerns raised by the reviewer.
>
> [1] Ibrahim Mansour I Alabdulmohsin, Behnam Neyshabur, and Xiaohua Zhai. Revisiting neural scaling laws in language and vision. In NeurIPS 2022, 2022.
>
>
> > 2) "As shown in the numerical results in the appendix, it seems in many cases the functional does not provide good prediction to the green points even though those points only span less than one order of magnitude in the horizontal axis."
>
> The subset of the plots in the appendix (e.g. subset of Caltech101) in which our functional form yields unsatisfactory extrapolations has unsatisfactory extrapolations due to the maximum (along the x-axis) of the black point used for fitting being near or before a break; this is accentuated by not having enough points for fitting for the SciPy fitter to be able to determine whether the break is an actual break or just noisy deviation. We discuss this limitation in Section 5.6 titled “The Limit of the Predictability of Scaling Behavior” in the paper. It is important to note that this limitation is not unique to our functional form. As can be seen in figure 21, 22, and 23 of https://arxiv.org/pdf/2209.06640v2.pdf#page=25 (Alabdulmohsin et al (2022)), every functional form (not just ours) of neural scaling behavior is unable to yield satisfactory extrapolations in such scenarios because the information needed to obtain a satisfactorily accurate extrapolation simply is not present in the data in such scenarios.
>
>
> > 3. The contributions are only marginally significant or novel.
>
> To the best of our (and apparently others) knowledge, no other paper has ever shown a functional form accurately modeling and extrapolating a set of scaling behaviors as diverse as this paper does.

---

> > ### Comment · Reviewer_TK8J · 2022-12-02
> > **Thanks for the response**
> >
> > Based on the response, the method does show good prediction capability on new network scales. Therefore, I raised my rating of the paper.
> >
> > There is one more question in which I am interested: many broken power law functionals can be used to fit a same set of data. For the results reported in the paper, did the authors do any model selection? If yes, what is the method and steps of model selection? (Perhaps related contents are included in the paper but I missed them.)

---

> > > ### Author Response · Authors · 2022-12-10
> > > **Reply 2 to reviewer TK8J**
> > >
> > > Thank you for your review and comments, we are glad that you found the contribution of our paper significant and interesting.
> > >
> > > > There is one more question in which I am interested: many broken power law functionals can be used to fit a same set of data. For the results reported in the paper, did the authors do any model selection? If yes, what is the method and steps of model selection? (Perhaps related contents are included in the paper but I missed them.)
> > >
> > > We select the model with lowest error on black points and don't use the held-out green points (i.e. the points used for evaluating extrapolation) for model selection.
> > >
> > > Except when stated otherwise in the paper, each plot contains a single break of a BNSL fit to black points which are smaller (along the x-axis) than the green points.
> > > As stated in the paper, Figure 2 left (Double Descent with model size on the x-axis) is an example of a plot that contains two breaks of a BNSL fit to black points which are smaller (along the x-axis) than the green points; we know that 2 breaks are necessary in this double descent scenario because (mathematically) there is no way to model 2 large non-monotonic transitions (present in the black points, not the green points that are held out to evaluate extrapolation) with a BNSL without using 2 breaks.
> > > Most of the time for most of the scaling behaviors that people care about in practical scenarios, the number of breaks needed to accurately model and extrapolate a scaling behavior is 1 break.

---

### Author Response · Authors · 2022-11-21
**Thank you for your reviews; new revision submitted; summary of changes below,**

We are very grateful for all the valuable suggestions from the reviewers. We have made several improvements to the paper. Here is a summary of these changes:

1) 100% of plots / figures now evaluate extrapolation. Originally, all plots except for reinforcement learning and double descent evaluated extrapolation.

2) We’ve added new figure called Figure 1 to help the reader visualize broken neural scaling laws.

3) Equation 1 has been recomposed into a more interpretable variant (that is mathematically equivalent to original equation 1).

4) All extrapolation results that we report use root mean squared log error as the metric for evaluating extrapolation.

5) The authors of Alabdulmohsin et al, 2022 [1] have since sent us new results for M1, M2, M3, and M4 in which “the number of iterations” used for fitting are increased. We now use these new results and Table 2 changes to:

....... Domain ............................................ |.. M1 ↑   |.. M2 ↑  |....  M3 ↑  |....  M4 ↑  |  BNSL ↑

....... Downstream Image Classification | 2.78% | 5.56% | 13.89% | 13.89% | 63.89%

....... Language ........................................ | 10%   ...|  10%  ....|   25%    .....|   0%    .......|     55%

6) Proof concerning M4 expressing inflection point is corrected.

7) We now additionally include multi-agent reinforcement learning extrapolation results in Figure 3.

8) The discussion is more detailed now in Section 5.6 titled "The Limit of the Predictability of Scaling Behavior".

9) On page 5 of revision, we accidentally added a typo paragraph that says "the manner that is Pareto optimal with respect to the performance evaluation metric on the y-axis (downstream accuracy in this case)." We will remove this typo paragraph for a camera ready version.

10) In Appendix A.1 and A.2, we’ve added more mathematical analysis about equation 1.  This includes a mathematical analysis of equation 1 in log-log space, revealing that it resembles a sum of softplus functions, which explains the (approximately) piece-wise linear shape shown in Figure 1.  We also believe this helps motivate equation 1 as a natural extension of the power-laws functions explored in previous work.

11) In Appendix A.7, we now additionally show that BNSL yields accurate extrapolations of the scaling behavior of diffusion generative models of images when amount of compute used for (pre-)training is on the x-axis and compute is scaled in the manner that is Pareto optimal with respect to the performance evaluation metric on the y-axis (negative log-likelihood (NLL) and Frechet inception distance (FID) score in this case).

12) In Appendix A.8, we now additionally show that BNSL yields accurate extrapolations of scaling behavior when data is pruned Pareto optimally (such that each point along the x-axis uses the subset of the dataset that yields the best performance (y-axis value) for that dataset size (x-axis value)).

13) In Appendix A.9, we additionally show that BNSL yields accurate extrapolations of scaling behavior when upstream performance is on the x-axis and downstream performance is on the y-axis.

14) In Appendix A.10, we additionally show that BNSL accurately extrapolates to scales that are order(s) of magnitude larger than the maximum (along the x-axis) of the points used for fitting.

15) In Appendix A.12, we now additionally include all plots of fits and extrapolations of M1, M2, M3, M4, and BNSL (overlaid on the same plot) on each of the tasks from Scaling Laws Benchmark of Alabdulmohsin et al, 2022 [1].


[1] Ibrahim Mansour I Alabdulmohsin, Behnam Neyshabur, and Xiaohua Zhai. Revisiting neural scaling laws in language and vision. In NeurIPS 2022, 2022.

---

### Author Response · Authors · 2023-05-17
**See arXiv version for longer version of this "Broken Neural Scaling Laws" paper with more results:   https://arxiv.org/abs/2210.14891**

See arXiv version for longer version of this "Broken Neural Scaling Laws" paper with more results:

https://arxiv.org/abs/2210.14891

https://arxiv.org/pdf/2210.14891.pdf

---

### Decision · Program_Chairs · 2023-01-20

**Decision:**

Accept: poster

**Justification For Why Not Higher Score:**

The paper has some serious shortcomings and limitations that must be iterated before becoming of higher impact. Details are provided in the meta-review summary.

**Justification For Why Not Lower Score:**

The work can potentially attracts constructive discussions at the conference. Details are provided in the meta-review summary.

**Metareview: Summary, Strengths And Weaknesses:**

A new neural scaling law is proposed that can predict and extrapolate to high-order scale regimes.

Strengths:
- Extensive empirical evidence is provided for the law
- The method is compared appropriately with the existing power-law models.
- The paper is very easy to follow and is very well organized.

Weaknesses:
- In almost all experimental results the [Alabdulmohsin et al. 2022] (M4) results seem to be as good on predictive performance as the method proposed.
- The limitation raised by the reviewers related to the inaccurate results for example Caltech101, is not negligible.
- Having a lot more hyperparameters in the scaling law as opposed to previous work is not ideal in this space, as performing ground truth experiments could be extremely expensive.

The paper is borderline, as there are multiple open concerns (some outlined above) that must be addressed about this new scaling law. However, based on my discussions with the reviewers, I believe that the paper can attract some useful discussions at the conference on neural scaling laws and, thus, vote for acceptance.

P.S. The claims must be toned down; Especially in the abstract and the intro as there are evident limitations observed where the introduced law failed.

**Note From Pc:**

if the above contains the word "oral" or "spotlight" please see: "oral" presentation means -> notable-top-5% and "spotlight" means -> notable-top-25%. As stated in our emails, we are disassociating presentation type from AC recommendations

**Summary Of Ac-Reviewer Meeting:**

We hold a zoom meeting with a sub-group of reviewers about this paper.

Here are the meeting notes:
- The nature of the work is very empirical and hard to assess formally. (This is totally fine, as work on scaling law is naturally empirical)
- The claims must be toned down; Especially in the abstract and the intro as there are evident limitations observed where the introduced law failed. (This is an important shortcoming of the work)
- The number of hyperparameters is high. (Could be managed with a few expert interventions)
- The authors did a great job improving their manuscript during the discussion period.